# RudLOV is an optically synchronized cargo transport method revealing unexpected effects of dynasore

Tatsuya Tago [1], Takumi Ogawa[1], Yumi Goto[2], Kiminori Toyooka [2], Takuro Tojima [3], Akihiko Nakano [3], Takunori Satoh [1✉] & Akiko K Satoh [1✉]

## Abstract

**Live imaging of secretory cargoes is a powerful method for understanding the mechanisms of membrane trafficking. Inducing the synchronous release of cargoes from an organelle is key for enhancing microscopic observation. We developed an optical cargo-releasing method, 'retention using dark state of LOV2' (RudLOV), which enables precise spatial, temporal, and quantity control during cargo release. A limited amount of cargo-release using RudLOV is able to visualize cargo cisternal-movement and cargo-specific exit sites on the Golgi/trans-Golgi network. Moreover, by controlling the timing of cargo-release using RudLOV, we reveal the canonical and non-canonical effects of the well-known dynamin inhibitor dynasore, which inhibits early- but not late-Golgi transport and exits from the trans-Golgi network where dynamin-2 is active. Accumulation of COPI vesicles at the cis-side of the Golgi stacks in dynasore-treated cells suggests that dynasore targets COPI-uncoating/tethering/fusion machinery in the early-Golgi cisternae or endoplasmic reticulum but not in the late-Golgi cisternae. These results provide insight into the cisternal maturation of Golgi stacks.**

**Keywords** Cisternal Maturation; Dynasore; Live Imaging; RudLOV; *trans*-Golgi Network
**Subject Categories** Membranes & Trafficking; Methods & Resources

## Introduction

Most membrane and secretory proteins are synthesized in the endoplasmic reticulum (ER), transported to the Golgi apparatus, and then travel to their destinations. Several components involved in these processes in various secretory pathways have been identified and characterized biochemically and structurally. Recent advances in optical and electron microscopy revealed new aspects of membrane interactions and cargo transfer (Kurokawa et al, 2014;

Shomron et al, 2021; Solinger et al, 2020; Weigel et al, 2021). Tracking secretory cargoes is a key technique in microscopic analyses and requires pulse labeling or the synchronous release of cargo proteins for clear visualization (Farr et al, 2009; Kurokawa et al, 2019; Lippincott-Schwartz et al, 2000; Presley et al, 1997).

Accordingly, several regulatory secretory cargoes and cargo retention and release systems have been developed (Boncompain et al, 2012; Bourke et al, 2021; Casler et al, 2020). The most widely used method is retention using selective hooks (RUSH) system, which uses streptavidin, streptavidin-binding peptide (SBP), and biotin (Boncompain et al, 2012). Streptavidin tagged an organelle-specific retention signal that traps SBP-tagged cargo proteins. Addition of biotin to the medium triggers the synchronous release of SBP-tagged cargo proteins from the organelle because biotin occupies the SBP-binding sites of streptavidin with high affinity. Cargo proteins fused to fluorescent proteins or tags allow for live imaging of cargoes during the journey from trapped organelle to the destination. However, the RUSH system is limited by its lack of spatial and quantity control for releasing cargos; moreover, supplying biotin to specific areas of cells is difficult, and stopping cargo release by washing out the biotin is nearly impossible because of the strong affinity of biotin for streptavidin, which results in prolonged cargo release. Light-triggered protein secretion systems allow for more precise control of cargo secretion. The first such system used a plant photoreceptor protein, UVR8, which forms a photolabile homodimer (Chen et al, 2013). Multiple UVR8-fused cargo proteins are sequestered in the ER, and a brief pulse of light triggers forward cargo trafficking through the secretory pathway to the plasma membrane. The rapid release of the cargo from the ER is robust, but the use of UV light (~310 nm) can damage cells. The same group recently developed another light-induced system, the zapalog-mediated ER trap (Bourke et al, 2021), in which milder 405-nm light illumination is used. In both systems, illumination can be precisely controlled at the single-cell or subcellular domain levels using the same microscope used for subsequent live imaging; cargo release can be triggered with exceptional spatial and temporal control. Bourke et al visualized and compared the trafficking pathways of synaptic proteins in neurons after the release of these proteins from the ER located in the central (cell body) and local (dendritic) regions. One limitation of this method is that a

[1]Program of Life and Environmental Science, Graduate School of Integral Science for Life, Hiroshima University, 1-7-1 Kagamiyama, Higashi-Hiroshima, Hiroshima 739-8521, Japan. [2]Technology Platform Division, Mass Spectrometry and Microscopy Unit, RIKEN Center for Sustainable Resource Science, Yokohama, Kanagawa 230-0045, Japan. [3]Live Cell Super-Resolution Imaging Research Team, RIKEN Center for Advanced Photonics, 2-1 Hirosawa, Wako, Saitama 351-0198, Japan. ✉E-mail: tsatoh3@hiroshima-u.ac.jp; aksatoh@hiroshima-u.ac.jp

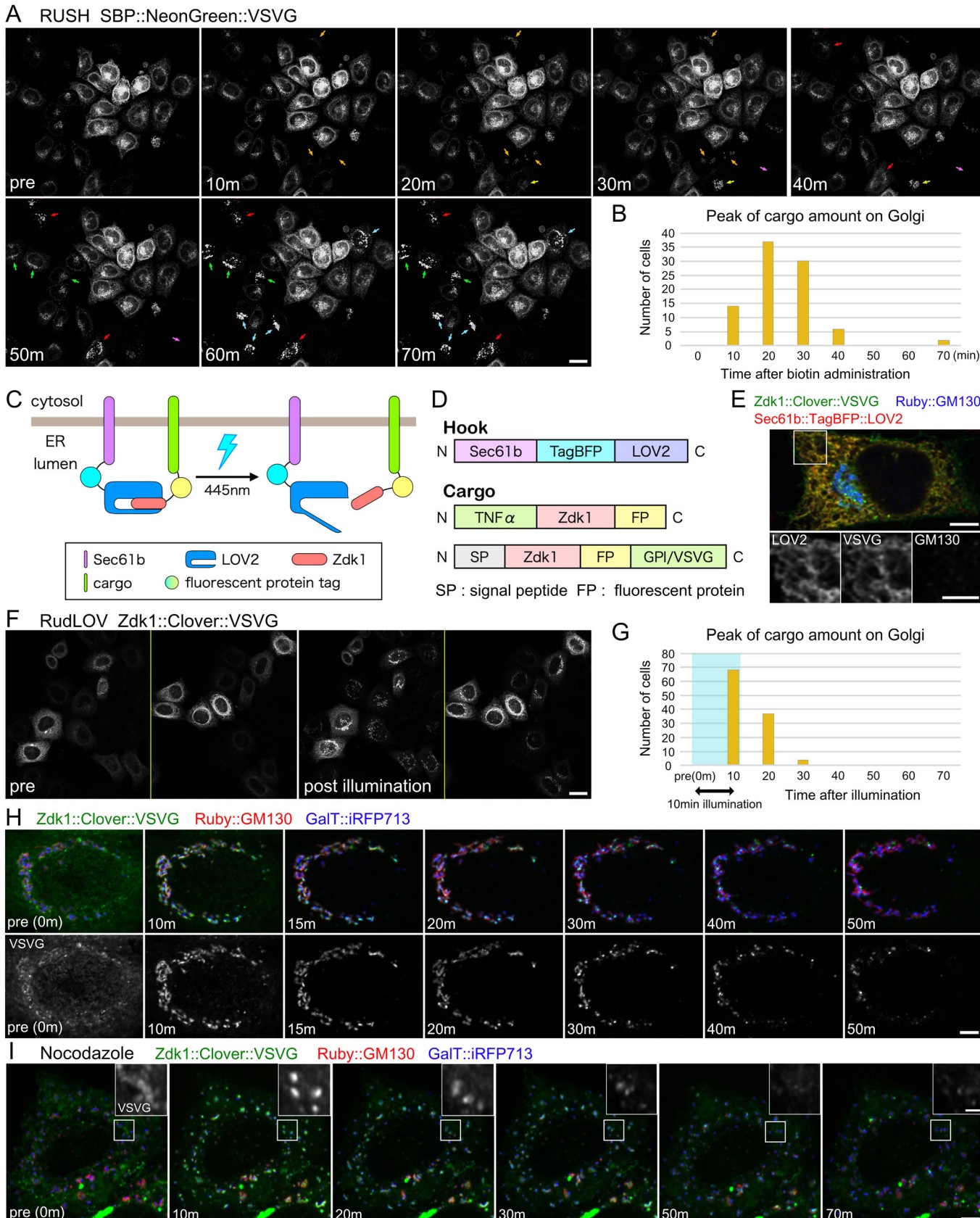

**A** RUSH SBP::NeonGreen::VSVG

pre | 10m | 20m | 30m | 40m
50m | 60m | 70m

**B** Peak of cargo amount on Golgi

**C** cytosol / ER lumen / 445nm

Sec61b / LOV2 / Zdk1 / cargo / fluorescent protein tag

**D**
Hook
N | Sec61b | TagBFP | LOV2 | C

Cargo
N | TNFα | Zdk1 | FP | C
N | SP | Zdk1 | FP | GPI/VSVG | C

SP : signal peptide  FP : fluorescent protein

**E** Zdk1::Clover::VSVG  Ruby::GM130
Sec61b::TagBFP::LOV2

LOV2 | VSVG | GM130

**F** RudLOV Zdk1::Clover::VSVG
pre | post illumination

**G** Peak of cargo amount on Golgi
pre(0m) 10 20 30 40 50 60 70
10min illumination  Time after illumination

**H** Zdk1::Clover::VSVG  Ruby::GM130  GalT::iRFP713
pre (0m) | 10m | 15m | 20m | 30m | 40m | 50m
VSVG
pre (0m) | 10m | 15m | 20m | 30m | 40m | 50m

**I** Nocodazole  Zdk1::Clover::VSVG  Ruby::GM130  GalT::iRFP713
pre (0m) | VSVG 10m | 20m | 30m | 50m | 70m

**Figure 1. RudLOV enables highly synchronized cargo-release.**

(A) SBP::NeonGreen::VSVG localizations before (left) and at 10, 20, 30, 40, 50, 60, and 70 min after administration of 100 μM of biotin in the RUSH system. Arrows indicate cells with accumulated SBP::NeonGreen::VSVG on the Golgi apparatus. (B) Plot of cell counts showing peak amount of cargo (VSVG) on the Golgi apparatus at each time point after biotin administration in RUSH system. (C, D) Schematic of the RudLOV method, and constructs of hook and cargoes. The N-terminal of LOV2 is fused with human Sec61b and fluorescent protein (TagBFP) to localize on the endoplasmic reticulum (ER). The cargoes and fluorescent proteins (Scarlet, Clover, or NeonGreen) are fused at either the N- or C-terminus of Zdk1, which binds to LOV2 in the dark and detaches on illumination at 445 nm. (E) Localization of the hook Sec61b::TagBFP2::LOV2 (red) and cargo Zdk1::Clover::VSVG (green) under dark conditions. Golgi apparatus is marked by GM130 (blue). (F) Zdk1::Clover::VSVG localizations before (left) and after (right) illumination at 445 nm using the RudLOV system. (G) Cell counts showing peak amount of cargo (VSVG) on the Golgi apparatus at each time point after the start of 10 min-illumination (blue) in RudLOV. (H, I) Localization of Zdk1::Clover::VSVG before and at 10, 15, 20, 30, 40, 50, and 70 min after onset of illumination at 445 nm using the RudLOV system in untreated (H) and nocodazole-treated cells (I). Inset in (I) shows a magnified image of Zdk1::Clover::VSVG. Data in (A), (E), (F), (H) and (I) are representative of more than three replicates. Scale bars: 20 μm (A), 5 μm (E), 2 μm (inset of E), 20 μm (F), 5 μm (H, I), and 2 μm (inset of I).

continuous supply of relatively expensive zapalogs is required after transfection or expression of cargo proteins to trap the cargo within the ER. In addition, the limited permeability of the chemicals makes it difficult to apply this method to whole animals and tissues. Furthermore, the 405 nm light required to degrade zapalogs damage cells, although not to the same extent as the damage caused by UV light.

In the present study, we report a light-induced cargo release method, named 'retention using dark state of LOV2' (RudLOV). RudLOV is based on LOVTRAP, which is an optogenetic approach for reversible light-induced protein dissociation (Wang et al, 2016). LOVTRAP utilizes two proteins: (1) LOV2 (light-oxygen-voltage), a photo-sensor domain from Avena sativa phototropin 1 and (2) a small artificial protein, Zdk1, which binds selectively to the dark state of LOV2. RudLOV utilizes LOV2 as the hook in the ER. The cargoes fused with Zdk1 selectively bind to the dark state of the LOV2 hook to be retained in the ER in the dark. Illumination at 443 nm triggers Zdk1-cargo release from the ER.

The ubiquitously expressed dynamin-2 plays an essential role in clathrin-mediated endocytosis, particularly in the scission of coated pits and vesicles from the plasma membrane (Ferguson and De Camilli, 2012). Several studies have reported that dynamin-2 is also required for post-Golgi transport; however, its precise roles in the Golgi stacks remain unclear (Cao et al, 2005; Jones et al, 1998; Kessels et al, 2006; Kreitzer et al, 2000; Salvarezza et al, 2009). Indeed, viable dynamin-1/2 double KO cells show accumulation of clathrin coated pits on the plasma membrane but not on the TGN or Golgi stacks (Ferguson et al, 2009). Thus, we investigated the effect of the well-known dynamin inhibitor dynasore (Macia et al, 2006) on cargo transport using RudLOV, and found an unexpected effect of dynasore on the *cis*-side in addition to the *trans*-side of Golgi stacks.

## Results and discussion

### RudLOV enables highly controlled cargo-release with spatial and temporal precision

The most widely used method for the synchronous release of cargo proteins is the RUSH system, which is convenient because cargo release is achieved simply by adding biotin to the medium. However, we observed an unexpected variation in the response of the cells to biotin when we used streptavidin::KDEL as a hook (Figs. 1A and EV1A). In 53% of cells, the cargo amount on the

Golgi apparatus peaked within 20 min after biotin administration (Fig. 1B). HeLa cells showing high levels of cargo expression after transient transfection with RUSH constructs tended to exhibit a longer delay in inducing cargo export from the ER after biotin administration (Fig. EV1B). This lack of temporal control in the RUSH system coupled with the lack of spatial and quantity controls for cargo release prompted us to develop a new method for the synchronous release of cargo proteins.

The schematics of the RudLOV method and constructs are shown in Fig. 1C,D. The LOV2 C-terminal helix Jα, which consists of the interface recognized by Zdk1, unwinds upon illumination. As the C-terminus of the LOV2 domain must be intact, instead of KDEL, the lumenal C-terminus of human Sec61b, which is a subunit of the translocon for ER-localization, was fused to LOV2. Cargoes and fluorescent proteins were tagged with Zdk1 at either the N- or C-terminus. In the dark, the cargoes were retained in the ER by the Sec61b::LOV2 hook (Fig. 1E). Upon illumination at 445 nm, the cargoes moved to the Golgi apparatus in all illuminated cells (Figs. 1F and EV1D). In 96% of cells, the cargo amount on the Golgi apparatus peaked within 20 min after the start of illumination (Fig. 1G). To avoid LOV2 activation during cargo detection, we used a 514 nm laser rather than a 488 nm laser to observe the Clover fluorescent proteins. To achieve seamless application of LOV2 activation and three-color imaging, we equipped an FV3000 confocal microscope with a quinta-band dichroic mirror for the 405/445/514/561/647 nm laser lines. This allowed for high-speed imaging without misalignment of the optical axis.

Three types of cargoes, glycosylphosphatidylinositol-anchored protein (GPI–AP), vesicular stomatitis virus protein G (VSVG), and tumor necrosis factor α (TNFα), were prepared (Fig. 1D). After 10 min of illumination, all three cargoes accumulated in the Golgi apparatus, and most exited from the Golgi apparatus within 50 min (Figs. 1H and EV1E,G). We also observed cargo movement in dispersed Golgi stacks after nocodazole treatment. All three types of cargoes accumulated in the Golgi apparatus after only 10 min of illumination, and most exited from the Golgi apparatus within 50 min (Figs. 1I and EV1F,H), similar to the results in nocodazole-untreated cells (Figs. 1H and EV1E,G). This result is consistent with that of a previous study in which transport was observed using the RUSH system and showed that microtubules are not strictly required for secretion (Fourriere et al, 2016).

Next, we investigated the ability of RudLOV to activate localized cargo export. We found that RudLOV activated cargo export in just

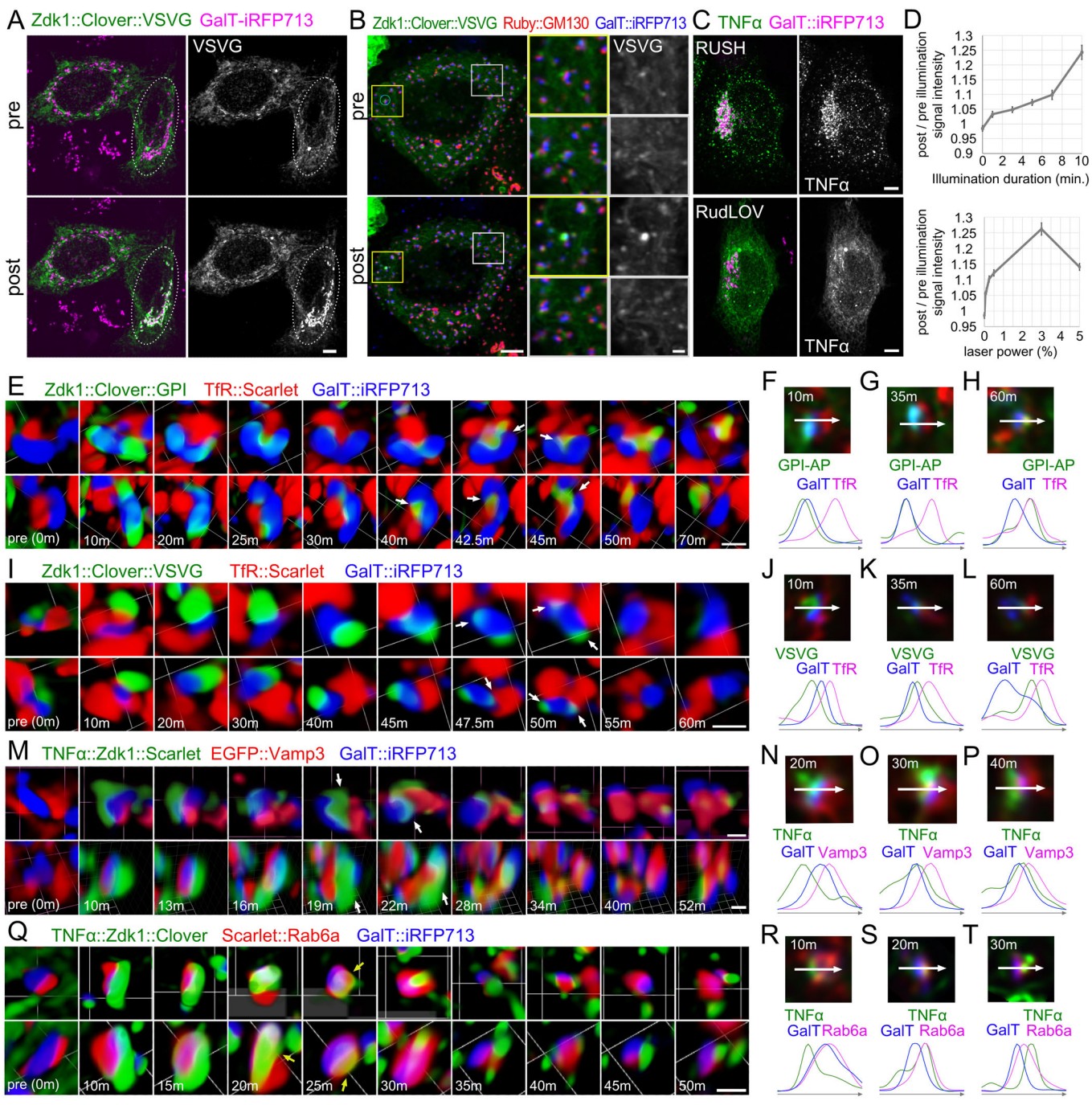

**A** Zdk1::Clover::VSVG  GalT-iRFP713

**B** Zdk1::Clover::VSVG  Ruby::GM130  GalT::iRFP713

**C** TNFα  GalT::iRFP713

**D**

**E** Zdk1::Clover::GPI  TfR::Scarlet  GalT::iRFP713

**F** GPI-AP  GalT TfR  **G** GPI-AP  GalT TfR  **H** GPI-AP  GalT TfR

**I** Zdk1::Clover::VSVG  TfR::Scarlet  GalT::iRFP713

**J** VSVG  GalT TfR  **K** VSVG  GalT TfR  **L** VSVG  GalT TfR

**M** TNFα::Zdk1::Scarlet  EGFP::Vamp3  GalT::iRFP713

**N** TNFα  GalT Vamp3  **O** TNFα  GalT Vamp3  **P** TNFα  GalT Vamp3

**Q** TNFα::Zdk1::Clover  Scarlet::Rab6a  GalT::iRFP713

**R** TNFα  GalT Rab6a  **S** TNFα  GalT Rab6a  **T** TNFα  GalT Rab6a

a single cell within a field (Fig. 2A) or within a single Golgi stack only, although some leaks were observed in neighboring Golgi stacks (Fig. 2B). Thus, RudLOV provides exceptional spatial and temporal secretion control. Notably, the RudLOV differed from the RUSH system in the manner of TNFα and VSVG localization before cargo release (Figs. 1A,F, 2C and EV1A,D). TNFα and VSVG are concentrated in the ER exit sites (ERES) in the RUSH system, but diffused throughout the ER in RudLOV. When biotin is not administrated, this ERES localization of VSVG is not changed for at least 10 min (Fig. EV1C). This difference in the localization of

ER-retained TNFα and VSVG may reflect differences in the ER-localization signals of the hooks, which is the KDEL in RUSH and Sec61b in RudLOV. Sec61b but not KDEL may have sufficient activity to exclude TNFα from ERES. Next, we investigated the ability of RudLOV to control the amount of cargo released. We estimated the relative amount of photo-released cargo from the peak intensity of cargo fluorescence in Golgi stacks. Illumination intensity was controlled by the duration or laser power. Stronger illumination resulted in higher peak intensity of cargo fluorescence in the Golgi stacks (Figs. 2D and EV2).

◀ **Figure 2. Localized activation of secretion using RudLOV.**

(A, B) Localization of the cargo Zdk1::Clover::VSVG approximately 10 min after illumination in untreated (A) and nocodazole-treated cells (B). The cargo Zdk1::Clover::VSVG is shown in green, and the *trans*-Golgi marker GalT::iRFP713 is shown in magenta (A) or blue (B). The right cell but not the left cell is illuminated (A). A single Golgi stack located in the middle of the yellow inset is illuminated (B). (C) Localization of the cargo TNFα::SBP::NeonGreen (upper) and TNFα::Zdk1::Clover (lower) before transport was triggered. The cargo is presented in green and the *trans*-Golgi marker GalT::iRFP713 is shown in magenta. (D) Relative amounts of cargo released by the different illumination levels. The peak intensities of cargo fluorescence on Golgi stacks are shown as the amount of cargo released. Intensity of illumination was controlled either by duration (upper plot) or laser power (lower plot). More than 32 cells are counted for each time point or laser power. Error bars are presented as mean ± SD. (E–T) Localization of the cargo Zdk1::Clover::GPI (E–H), Zdk1::Clover::VSVG (I–L), TNFα::Zdk1::Scarlet (M–P), and TNFα::Zdk1::Clover (Q–T) before (left) and after illumination in a single Golgi/RE unit in nocodazole-treated cells. One each of time lapse 3D volume containing a representative Golgi/RE unit is presented as pairs of time-series volumetrically rendered from two different view angles (E, I, M, Q). The time after illumination is shown in the bottom-left corner. Plots show signal intensities from the image on the left Golgi/RE unit at indicated time points after illumination (F–H, J–L, N–P, R–T). Signal intensity was measured along the arrow (representing 1.5 μm). Arrows indicate Zdk1::Clover::GPI is on RE (E) and Zdk1::Clover::VSVG is not on RE (I). Arrows indicate TNFα::Zdk1::Scarlet on the membrane between *trans*-Golgi cisternae and RE (M) and TNFα::Zdk1::Clover on the Rab6-positive compartment (Q). The cargo is shown in green, a *trans*-Golgi marker GalT::iRFP713 in blue, and the RE marker TfR::Scarlet (E–L), RE marker EGFP::Vamp3 (M–P), or early-TGN marker Scarlet::Rab6a (Q–T) in red. Data in (A–C), (E), (I), (M) and (Q) are representative of more than three replicates. Scale bars: 10 μm (A), 5 μm (B, C), 2 μm (inset in B), and 1 μm (E, I, M, Q).

## TNFα and VSVG accumulate in the membrane between *trans*-cisternae and Golgi-associated recycling endosomes before exiting the Golgi stacks

We previously observed a relationship between the Golgi stacks and recycling endosomes (RE) in nocodazole-treated cells, and described two types of REs: free RE and Golgi-associated RE (GA–RE). Most Golgi stacks are accompanied by GA–RE (Fujii et al, 2020a; Fujii et al, 2020c). Here, we refer to the Golgi stack/GA–RE complex as the Golgi/RE unit. We previously showed that GPI–AP is transported from the Golgi stack to the GA–RE before exiting the Golgi/RE unit, whereas VSVG did not. To understand the details of cargo transport within the Golgi/RE unit in nocodazole-treated HeLa cells, we visualized the movement of small amounts of cargo released under limited illumination. Prior to illumination, GPI–AP, VSVG, and TNFα were not detected around the Golgi stacks (Fig. 2E,I,M,Q left). After 5 min of illumination, GPI–AP was detected near the Golgi stack. Golgi stacks were accompanied by GA–RE during cargo secretion, and GPI–AP moved seamlessly from Galactosyltransferase (GalT) positive *trans*-Golgi cisternae to transferrin receptor (TfR or Vamp3)-marked GA–RE (Fig. 2E–H; Appendix Fig. S1A; Fig. EV3A). After 5 min of illumination, VSVG was also detected around the Golgi stack but did not enter the TfR-marked GA–RE even at 35 and 60 min after illumination (Fig. 2I–L; Appendix Fig. S1B). VSVG remained near the *trans*-Golgi cisternae and GA–RE until around 40 min (Fig. 2I arrow; Appendix Fig. S1B). This result is consistent with those of a previous study in which the RUSH system was used (Fujii et al, 2020b). Interestingly, TNFα appeared to force GA–RE away to create more space for it to remain in the gap (Fig. 2M arrow; Appendix Fig. S2A; Fig. EV3B) between the *trans*-Golgi cisternae and GA–RE after its exit from the *trans*-Golgi cisternae (Fig. 2M–P; Appendix Fig. S2A; Fig. EV3B). To address the identity of the TNFα-positive membrane in the gap, we visualized the Rab6-positive compartment. Rab6 was previously described to locate on early-TGN in budding yeast (Tojima et al, 2019) and on the boundary between trans-cisternae and TGN in mammalian cells (Tie et al, 2018). Here, TNFα seamlessly moved from the GalT-positive *trans*-Golgi cisternae to the Rab6-positive early-TGN (Fig. 2Q–T; Appendix Fig. S2B), and then passed the early-TGN within approximately 20 min (Fig. 2S). TNFα remained in an unidentified compartment attached to the Rab6-positive early-TGN until it exited the Golgi/RE unit (Fig. 2T). This unidentified compartment may be the late-TGN defined by clathrin and GGA in yeast (Tojima et al, 2019). Clathrin and GGA are also known to be localized on the punctate structures within TGN but not adjacent to *trans*-cisternae in mammalian cells (Tie et al, 2018). These results indicate that the ability of RudLOV to precisely control the amount and timing of cargo release is useful for understanding cargo movement.

## Dynasore inhibits early-stage intra-Golgi transport

Dynamin-2 is required for the formation of post-Golgi vesicles in the TGN (Cao et al, 2005; Jones et al, 1998; Kessels et al, 2006; Kreitzer et al, 2000; Salvarezza et al, 2009). A previous study showed that administration of dynasore to cells inhibited cargo transport on or around the Golgi apparatus; however, a detailed analysis was not performed (Weller et al, 2010). Hence, we used RudLOV to investigate the effects of dynasore on cargo transport. We observed cargo movement within the dispersed Golgi/RE unit in nocodazole-treated cells. Without dynasore, VSVG accumulated at the *cis*-side of the Golgi stacks immediately after illumination (defined as 10 min) but colocalized poorly with the *cis*-Golgi marker GM130. This suggests that the cargo did not reach the *cis*-cisternae within 10 min (Fig. 3A). However, 15 min after onset of illumination, VSVG was colocalized with GM130; at 20 min, some of the cargo began to reach the *trans*-Golgi cisternae marked by GalT::iRFP713. Furthermore, at 40 min, the cargo had completely passed through the *trans*-Golgi cisternae and gradually disappeared. Finally, after 130 min, no cargo were detected around the Golgi/RE units. Unexpectedly, when dynasore was administered 1 min before cargo photo-release, the transport of VSVG was stalled at the *cis*-side of the Golgi stacks (Fig. 3C). VSVG continuously colocalized strongly with GM130 for at least 55 min (Fig. 3D). However, 130 min after illumination, cargo was detected between GM130 and GalT (Fig. 3E), indicating that the cargo was slowly sent forward. Meanwhile, even at 60 min after administration of dynasore, the Golgi stacks strongly maintained their *cis*-*trans* polarity (Fig. 3B). We also observed the effects of dynasore on the transport of GPI–AP and TNFα. When dynasore was administered 1 min before cargo photo-release, both GPI–AP and TNFα continued to colocalize with GM130, even at 100 and 70 min after illumination, respectively, similar to the results observed for VSVG (Fig. 3F,G).

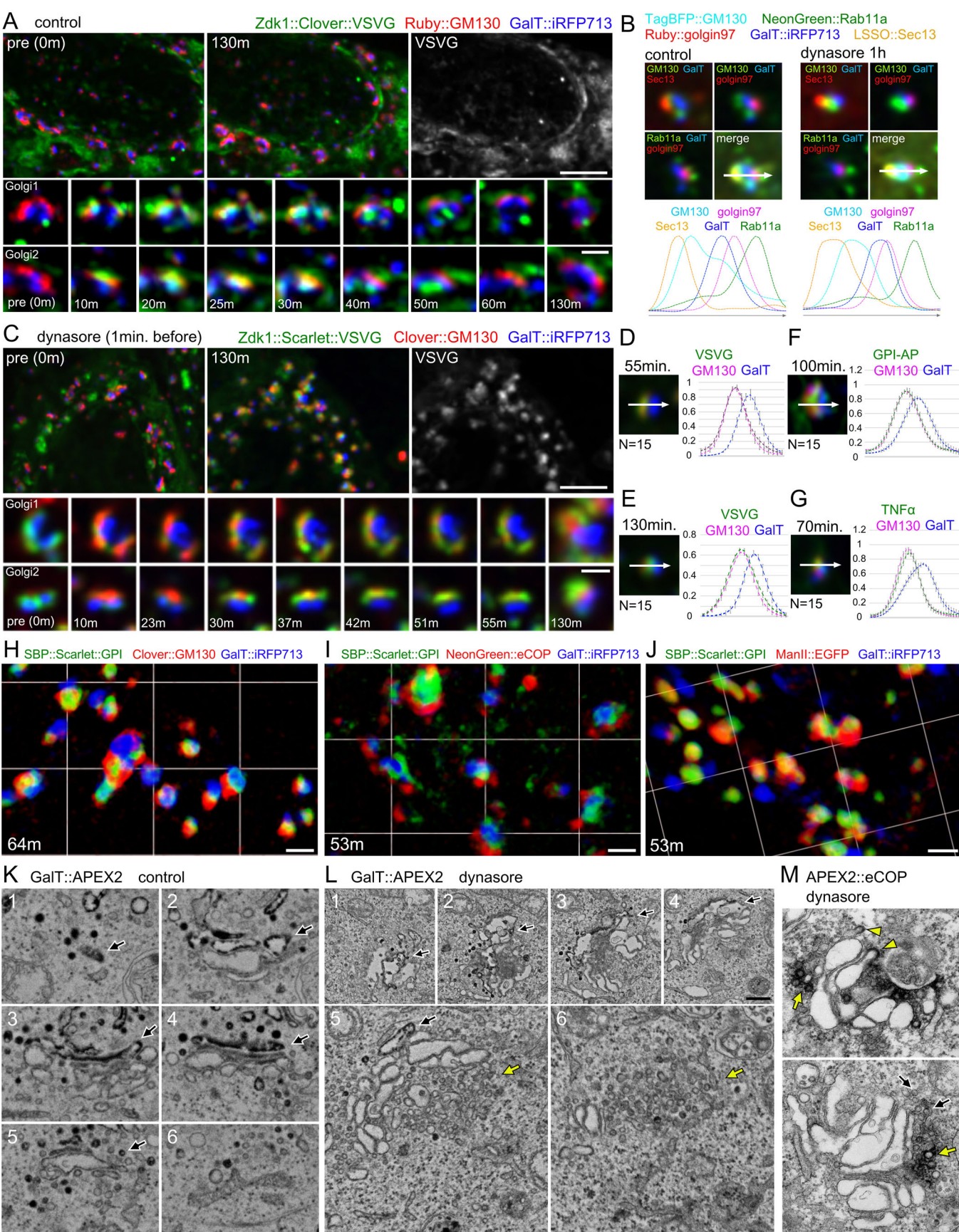

**Figure 3. Pre-administration of dynasore-inhibited cargo transport at the *cis*-side of the Golgi stacks.**

(A) Localization of Zdk1::Clover::VSVG before illumination (upper left) and 130 min after onset of illumination (upper middle and right) in untreated cells. The time-course of localization of Zdk1::Clover::VSVG before and after illumination of an untreated single Golgi stack (bottom). Zdk1::Clover::VSVG is shown in green, a *cis*-Golgi marker Ruby::GM130 in red, and a *trans*-Golgi marker GalT::iRFP713 in blue. (B) Upper panels show examples of quintuple-stained Golgi/RE units using the indicated fluorescent tagged proteins before and 1 h after administration of dynasore. Plots show signal intensities from the image on the upper panel. Signal intensity was measured along the arrow (representing 1.5 μm) in the inset. The graph shows the overlap between channels. (C) Localization of Zdk1::Scarlet::VSVG before illumination (upper left) and at 130 min after onset of illumination (upper middle and right) in cells in which dynasore was administered 1 min before cargo photo-release. The time-course of localization of Zdk1::Scarlet::VSVG before and after illumination of the dynasore-treated single Golgi stack (bottom). Zdk1::Scarlet::VSVG is shown in green, *cis*-Golgi marker Clover::GM130 in red, and *trans*-Golgi marker GalT::iRFP713 in blue. (D–G) Plots show the normalized means of 15 line profiles of Cargo, GM130 and GalT across the Golgi stack. The 15 line profiles were obtained from five Golgi stacks per cell, using three different cells. The left image shows the typical Golgi/RE unit at 55 min (D), 130 min (E), 100 min (F), and at 70 min (G) after onset of illumination. Signal intensity was measured along the arrow (representing 1.5 μm). The cargoes Zdk1::Scarlet::VSVG (D, E), Zdk1::Clover::GPI (F), and TNFα::Zdk1::Clover (G) are in green, *cis*-Golgi marker Clover/Ruby::GM130 in red, and *trans*-Golgi marker GalT::iRFP713 in blue. (H–J) Volumetrically-rendered images for localization of SBP::Scarlet::GPI (green) in dynasore-administered cells after biotin-addition observed using SCLIM. Clover::GM130 (H), NeonGreen::eCOP (I), or ManII::EGFP (J) is shown in red and *trans*-Golgi marker GalT::iRFP713 in blue. The time shown in the bottom-left corner is the time after biotin-addition. (K, L) Scanning electron micrographs of serial sections of a Golgi stack with 200 nm-interval in the cell after 1 h of incubation with (L) or without 100 μM dynasore (K). GalT::APEX2 visualized trans-Golgi cisternae and vesicles (black arrows). Yellow arrows indicate the accumulation of vesicles on the *cis*-side of the Golgi stack. (M) Transmission electron micrographs of Golgi stacks with APEX2::eCOP showing COPI budding profiles (yellow arrowheads) and vesicles (yellow arrows). Black arrows indicate vesicles without staining. Data in (A–C) and (H–J) are representative of more than three replicates. Scale bars: 5 μm (upper panel in A, C), 1 μm (lower panel in A, C), 2 μm (H–J), and 500 nm (K–M).

We examined the precise localization of cargoes using super-resolution confocal live imaging microscopy (SCLIM). Owing to the limitation of the light source equipped on SCLIM, we used the RUSH system for the transport assay rather than using RudLOV. We found that with 1 min prior administration of dynasore, the cargo was partially and fully colocalized with the *cis*-cisternae marker GM130 and *medial*-cisternae marker ManII at 64 and 53 min after biotin addition, respectively. However, these proteins did not colocalize with eCOP at 53 min after biotin addition (Fig. 3H–J). Thus, the cargo was located within the *cis*- and *medial*-cisternae rather than in the COPI vesicles. These results suggest that dynasore inhibits the maturation of *cis/medial*-Golgi cisternae to *trans*-Golgi cisternae.

Ultrastructural observation of dynasore-treated cells using serial section electron microscopy revealed that the Golgi stacks were accompanied by numerous vesicles (Figs. 3K–M and EV5A,B,D; Movie EV1). We counted the number of vesicles near Golgi stacks in dynasore treated and untreated cells (Fig. EV5C) and identified 810.2 (±248.2) and 326.8 (±83.4) vesicles near a Golgi stack, respectively. These vesicles (Fig. 3L yellow arrows) were mainly found on the opposite side to the GalT::APEX2-marked *trans*-Golgi cisternae (Fig. 3L, black arrows), suggesting that the vesicles localized around the *cis*-side of the Golgi stacks. Since some vesicles were positively marked by APEX2::eCOP (Fig. 3M, yellow arrows), these vesicles were likely to be COPI vesicles. These results suggest that dynasore inhibits the uncoating, tethering, or fusion of COPI vesicles to prevent the maturation of *cis/medial*-Golgi cisternae to *trans*-Golgi cisternae. Off target effects of dynasore on fluid-phase endocytosis and membrane ruffling has been reported (Park et al, 2013), but there is no report showing inhibition of early Golgi transport by dynasore, as we found here.

## Dynasore exerts a canonical inhibitory effect on cargo export from the TGN

The unexpected inhibition of early Golgi cargo transport by dynasore can mask its canonical inhibitory effect on dynamin-2 in the TGN. To prevent this effect, we delayed dynasore administration to slightly after cargo release and evaluated various intervals between the times of illumination and dynasore administration. Notably, administration of dynasore 14 min after onset of illumination allowed VSVG to pass through the *cis*- and *trans*-Golgi cisternae but inhibited export from the Golgi/RE units. VSVG remained at the supposed TGN on the *trans* side of the GalT::iRFP713 compartment for more than 130 min following illumination (Fig. 4A–C). Similar post-illumination administration was performed for GPI–AP and TNFα. We observed similar *cis* to *trans* migration of the cargo and inhibition of exit from the TGN (Fig. 4D,E). The distinct effects of pre- and post-illumination administration of dynasore were also observed in cells without nocodazole treatment (Fig. EV4A–D).

We also investigated the dose-dependency of pre-administered dynasore in terms of inhibitory effects on cargo transport at the *cis*-Golgi cisternae and TGN. Pre-administration of 12 or 25 μM dynasore inhibited the cargo exit from the TGN, but pre-administration of 50 or 100 μM dynasore inhibited cargo movement at the *cis*-Golgi cisternae (Fig. EV4E–L). The former is comparable to the IC50 value of dynasore to dynamin-1/2, but the latter is higher. To confirm dynamin-2 inhibition, we confirmed that 100 μM dynasore inhibited the uptake of transferrin (Fig. EV4M).

The time points of co-localization of VSVG with GM130 and GalT in dynasore-administered cells before and after illumination and in dynasore-untreated control cells were plotted (Fig. 4F). GM130/VSVG-colocalization peaked approximately 20 min after the start of illumination in untreated cells. GalT/VSVG co-localization gradually increased, peaking approximately 30 min after illumination, and then rapidly decreased. In pre-illumination dynasore-administered cells, GM130/VSVG-colocalization remained high 130 min after the start of illumination, and GalT/VSVG-colocalization was not elevated during the 130 min observation period. In post-illumination dynasore-administered cells, after temporal GM130/VSVG colocalization, GalT/VSVG co-localization reached a peak at 30 min and remained high until at least 60 min. These plots clearly indicate that intra-Golgi transport was inhibited by pre-illumination administration of dynasore at the early stage but not at the later stage.

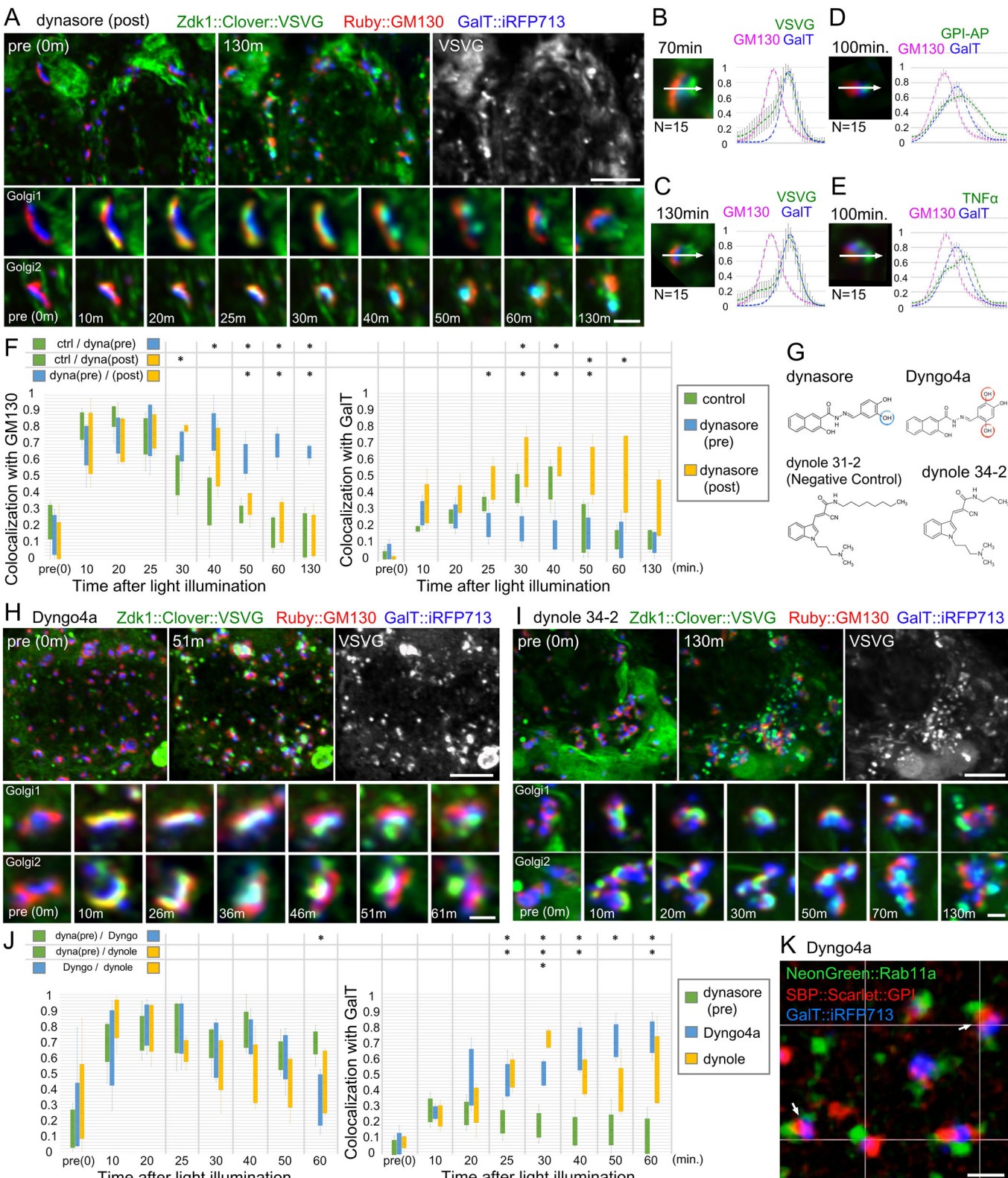

**Other dynamin inhibitors, Dyngo4a and dynole 34-2, inhibit cargo transport only at the TGN**

The inhibition of exit from the TGN agrees with the results of previous studies, which showed that dynamin-2 is required for forming post-

Golgi vesicles in the TGN (Cao et al, 2005; Jones et al, 1998; Kessels et al, 2006; Kreitzer et al, 2000; Salvarezza et al, 2009). However, inhibition of the early stages of transport in the Golgi stacks by dynasore was unexpected. Thus, we examined the effects of other dynamin inhibitors, such as Dyngo4a, dynole 34-2 (dynole), and

◄

**Figure 4. Post-administration of dynasore 4 min after illumination inhibits cargo transport at the *trans*-Golgi network (TGN).**

(A) Localization of Zdk1::Clover::VSVG before illumination (upper left) and 130 min after illumination (upper middle and right) in cells post-administered with dynasore at 14 min after onset of illumination. The time-course of localization of Zdk1::Clover::VSVG before and after onset of illumination in a single Golgi stack (bottom) with post-administered dynasore is shown. Zdk1::Clover::VSVG is shown in green, *cis*-Golgi marker Ruby::GM130 in red, and *trans*-Golgi marker GalT::iRFP713 in blue. (B–E) Plots show the normalized means of 15 line profiles of the cargo, GM130, and GalT across the Golgi stack. The 15 line profiles were obtained from five Golgi stacks per cell, using three different cells. The left image shows the typical Golgi/RE unit at 70 min (B), 130 min (C), 100 min (D), and 100 min (E) after onset of illumination of cells with post-administration of dynasore 4 min after illumination. Signal intensity was measured along the region indicated by the arrow (representing 1.5 μm). The cargo, Zdk1::Clover::VSVG (B, C), Zdk1::Clover::GPI (D), and TNFα::Zdk1::Clover (E) are shown in green, *cis*-Golgi marker Ruby::GM130 in red, and *trans*-Golgi marker GalT::iRFP713 in blue. (F) Plots show the colocalization of the cargo with Ruby::GM130 (left) and GalT::iRFP713 (right) in cells that are untreated (green), pre-administered (blue), and post-administered (yellow) with dynasore. Error bars indicate the standard deviation from 5 Golgi stacks. Results of Steel-Dwass test for each combination of samples are shown in rows on top (*P < 0.05). Exact P values (to 4 decimal points) for GM130 of (F) ctrl vs dyna(pre) 0.4519(pre), 0.4488(10), 0.5870(20), 0.9879(25), 0.2222(30), 0.0212(40), 0.0198(50), 0.0221(60), 0.0194(130), ctrl vs dyna(post) 0.5227(pre), 0.7145(10), 0.1345(20), 1.0000 (25), 0.0220(30), 0.3264(40), 0.4537(50), 0.8352(60), 1.0000(130) and dyna(pre) vs dyna(post) 0.7970(pre) 0.9881(10), 1.0000(20), 0.9879(25), 0.0810(30), 0.7142(40), 0.0198(50), 0.0221(60), 0.0194(130). Exact P values (to 4 decimal points) for GalT of (F) ctrl vs dyna(pre) 0.9928(pre), 0.2227(10), 0.9869(20), 0.2234(25), 0.0215(30), 0.0427(40), 0.9883(50), 1.0000(60), 0.8405(130), ctrl vs dyna(post) 0.5230(pre), 0.1387(10), 0.2242(20), 0.2234 (25), 0.1393(30), 0.2259(40), 0.0433(50), 0.0428(60), 0.2298(130) and dyna(pre) vs dyna(post) 0.3906(pre) 0.8410(10), 0.3239(20), 0.0440(25), 0.0215(30), 0.0207(40), 0.0433(50), 0.0832(60), 0.2298(130). Error bars are presented as mean ± SD. (G) Structure formulas of four compounds used in this work. Dyngo4a has an extra hydroxy group compared with dynasore. (H, I) Localization of Zdk1::Clover::VSVG before illumination (upper left) and 130 min after onset of illumination (upper middle and right) in cells pre-administered with Dyngo4a (H) or dynole 34-2 (I). The time-course of localization of Zdk1::Clover::VSVG before and after illumination in Dyngo4a-treated or (H) or dynole 34-2-treated (I) single Golgi stack (bottom). Zdk1::Clover::VSVG is shown in green, *cis*-Golgi marker Ruby::GM130 in red, and *trans*-Golgi marker GalT::iRFP713 in blue. (J) Plots showing colocalization of the cargo with Ruby::GM130 (left) and GalT::iRFP713 (right) in cells pre-administered with dynasore (green), Dyngo4a (blue), or dynole 34-2 (yellow). Error bars indicate the standard deviation from 5 Golgi stacks. Results of Steel-Dwass test for each combination of samples are shown in rows on top (*P < 0.05). Exact P values (to 4 decimal points) for GM130 of (J) dyna(pre) vs dynole 0.5099(pre), 0.9879(10), 0.5773(20), 0.8419(25), 1.0000 (30), 0.9873(40), 0.4472(50), 0.0235(60), dyna(pre) vs Dyngo 1.0000(pre), 0.0197(10), 0.8336(20), 0.3327(25), 0.4570(30), 0.2279(40), 0.3233(50), 0.0812(60), dynole vs Dyngo 0.5099 (pre), 0.0768(10), 1.0000 (20), 0.5875(25), 1.0000 (30), 0.2279(40), 0.0773(50), 0.8421(60). Exact P values (to 4 decimal points) for GalT of (J) dyna(pre) vs dynole 0.8237(pre), 0.9872(10), 0.5773(20), 0.0442(25), 0.0211 (30), 0.0222(40), 0.0212(50), 0.0217(60), dyna(pre) vs Dyngo 0.6243(pre), 0.8398(10), 0.8336(20), 0.0213(25), 0.0211(30), 0.0222(40), 0.0790(50), 0.0217(60), dynole vs Dyngo 0.7917 (pre), 0.8398(10), 1.0000 (20), 0.8409(25), 0.0211 (30), 0.2192(40), 0.1359(50), 0.3261(60). Error bars are presented as mean ± SD. (K) Volumetrically-rendered images for localization of SBP::Scarlet::GPI (red) in Dyngo4a-administered cells 35 min after biotin-addition observed using SCLIM. The RE marker NeonGreen::Rab11a is shown in green and *trans*-Golgi marker GalT::iRFP713 is in blue. Arrows indicate the localization of SBP::Scarlet::GPI on the membrane between *trans*-Golgi cisternae and RE. Data in (A), (H) and (I) are representative of more than three replicates. Scale bars: 5 μm (upper panel in A, H, I), 1 μm (lower panel in A, H, I), and 2 μm (K).

dynole31-2 (dynole negative control [NC]) (Fig. 4G; (McCluskey et al, 2013; Robertson et al, 2014; Stallaert et al, 2018). Pre-illumination administration of either Dyngo4a or dynole inhibited cargo export from the TGN but did not strongly affect early Golgi transport (Fig. 4H,I). Pre-administration of dynole NC did not affect cargo transport at all (Fig. EV4N). The time-course plots of GM130/VSVG- or GalT/VSVG-colocalization showed that pre-illumination administration of Dyngo4a or dynole allowed cargo migration from GM130-positive cisternae to GalT-positive cisternae, whereas exit from Golgi/RE units was inhibited (Fig. 4J). As GPI–AP is transported to the GA–RE after passing through GalT-positive *trans*-cisternae in untreated cells, we investigated whether GPI–AP reached the GA–RE in Dyngo4a-treated cells. Observations using SCLIM clearly revealed that GPI–AP was separated from the RE marker Rab11a (Fig. 4K). Thus, Dyngo4a inhibits transport from the Golgi stacks to the GA–RE. We observed dynole, dynole NC (control), and Dyngo4a treated cells with eCOP-APEX2 by transmission electron microscopy (TEM; Fig. EV5D–G). Some COPI vesicles were observed in these Golgi stacks, but the average number of vesicles counted in single TEM sections was increased only in dynasore treated cells (Fig. EV5H).

These effects of Dyngo4a and dynole agree with the reports that dynamin-2 is essential for post-Golgi transport but not for early Golgi transport (Cao et al, 2005; Jones et al, 1998; Kessels et al, 2006; Kreitzer et al, 2000; Salvarezza et al, 2009). Thus, dynasore likely inhibits dynamin-2 at the TGN, resulting in inhibition of post-Golgi transport; additionally, it inhibits another target at the *cis*-side of Golgi stacks, resulting in defects in the maturation of *cis/medial*-cisternae to *trans*-cisternae. Cisternal maturation of the Golgi stack is driven by retrograde trafficking of Golgi-resident proteins by COPI vesicles (Emr et al, 2009; Ishii et al, 2016;

Papanikou et al, 2015; Papanikou and Glick, 2014). Together with the electron micrographs showing COPI vesicle accumulation in dynasore-administered cells, the unidentified target of dynasore at the *cis*-side of the Golgi stacks may be involved in uncoating/tethering/fusion of COPI vesicles.

Our results indicate that cargo transport in the early (*cis/medial* to *trans*), but not late (*trans* to TGN), stages is inhibited by dynasore. Thus, dynasore may inhibit the factors involved in early cisternal maturation. Interestingly, the tether and SNAREs involved in the fusion of COPI vesicles with the ER differ from those involved in the fusion of COPI vesicles with Golgi cisternae; the NRZ complex and SEC20/STX18/USE1/SEC22 are associated with the former, whereas the COG complex and STX5/GOS28/YKT6/GS15 are associated with the latter (Hong and Lev, 2014; Ren et al, 2009; Zink et al, 2009). In addition, there are three types of ARFs with different localizations on the *cis*-side of Golgi stacks: ARF4/5 is mainly localized on the ERGIC, while ARF1 is mainly localized on the *cis*-Golgi cisternae and TGN (Chun et al, 2008; Duijsings et al, 2009; Wong-Dilworth et al, 2023). The function of ARF4/5 is not fully understood but they have the activity for COPI assembly at least in vitro (Adolf et al, 2019; Popoff et al, 2011). ARF1 is known to be required for not only assembly, but also scission and uncoating of COPI vesicles; scission of COPI vesicles from Golgi-cisternae requires ARF1 dimers (Beck et al, 2011; Diestelkoetter-Bachert et al, 2020), and GTP-hydrolysis on ARF1 is pre-requisite for COPI uncoating (Tanigawa et al, 1993; Zink et al, 2009). Recent studies have indicated that retrograde trafficking to the ER is impaired in ARF4 knockout but not in ARF1 knockout cells (Pennauer et al, 2022). Yeast cells house two ARFs, namely, Arf1 and Arf2, and it is difficult to tell which yeast Arf is the functional

counterpart of mammalian ARF4/5 based on homology analysis. Notably, the maturation of early Golgi cisternae becomes slower and less frequent in Arf1-deficient yeasts, whereas the maturation of late Golgi cisternae does not change (Bhave et al, 2014). Thus, dynasore may inhibit GTP-hydrolysis on ARF4/5 or COPI-uncoating/tethering/fusion machinery in the early-Golgi cisternae or ER, but not in the late-Golgi cisternae. Future studies are required to identify dynasore targets on the *cis*-side of the Golgi stacks to reveal the underlying mechanism of cisternal maturation.

## Conclusion

We developed a new cargo releasing method, RudLOV, which has three exceptional control abilities: spatial, temporal, and quantitative control of cargo release. In addition, the use of the relatively longer wavelengths of light (445–488 nm) reduces damage to the cells compared with the previously reported UVR8-based light trigger method (310 nm) or zapalog-mediated ER trap (405 nm) (Bourke et al, 2021; Chen et al, 2013). Quantitative control of cargo release by RudLOV enables visualization of cisternal movement of the cargo and the cargo-specific exit site on the Golgi/TGN. Moreover, precise temporal control ability of cargo release by RudLOV can be used to dissect canonical and non-canonical effects of the well-known dynamin inhibitor dynasore. LOV2 can be activated by 445 or 488 nm light, offering damage-less application of light-induced cargo export. Moreover, the simplicity of RudLOV without chemical application allows for cargo export in cells in situ or in whole animals. These exceptional control abilities and convenience of use demonstrate the potential of using RudLOV for sophisticated observation of cargo trafficking.

## Methods

### Reagents and tools table

| Reagent/Resource | Reference or Source | Identifier or Catalog Number |
|---|---|---|
| **Experimental models** | | |
| HeLa cells (*H. sapiens*) | ATCC | CCL-2 |
| **Recombinant DNA** | | |
| CMV-ManII-EGFP | This work | |
| CMV-2xCv-GM130 | This work | |
| CMV-2xRb-GM130 | This work | |
| CMV-TfR-2xSca | This work | |
| pEGFP VAMP3 | Addgene | 42310 |
| CT7-2xSca-Rab6a | This work | |
| CT7-2xNG-COPe | This work | |
| pEBP-Sec61b-mTagBFP2-LL-LOV2 | This work | |
| CMV-SP-Zdk-2xCv-GPI | This work | |
| CMV-SP-Zdk-2xCv-VSVG | This work | |
| CMV-SP-Zdk-2xSca-VSVG | This work | |

| Reagent/Resource | Reference or Source | Identifier or Catalog Number |
|---|---|---|
| CMV-TNF-Zdk-2xCv | This work | |
| CMV-TNF-Zdk-2xSca | This work | |
| Str-KDEL_SBP-2xSca-GPI | This work | |
| Str-KDEL_SBP-2NG-VSVG | This work | |
| **Chemicals, Enzymes and other reagents** | | |
| D-MEM(High Glucose) | FUJIFILM Wako Chemicals | 045-30285 |
| Penicillin-Streptomycin-L-Glutamine Solution (×100) | FUJIFILM Wako Chemicals | 161-23201 |
| 100 mmol/L Sodium Pyruvate | FUJIFILM Wako Chemicals | 190-14881 |
| 1 mol/l-HEPES Buffer | Nacalai Tesque | 17557-94 |
| Fetal Bovine Serum | Nichirei | 174012 |
| Dynasore | Cayman Chemicals | 14062 |
| Nocodazole | Cayman Chemicals | 13857 |
| Cycloheximide | Cayman Chemicals | 14126 |
| Dynole 34-2 | Abcam | 120464 |
| Dynole 31-2 | Abcam | 120474 |
| Dyngo-4a | Cayman Chemicals | 29479 |
| Biotin | FUJIFILM Wako Chemicals | 029-08713 |
| Biliverdine | TRC | B386400 |
| ibidi® μ-Slide 8-well | ibidi | 80826 |
| JetPrime | Polyplus | 101000015 |
| JetOptimus | Polyplus | 101000006 |
| G418 sulfate | FUJIFILM Wako Chemicals | 078-05961 |
| Puromycin Dihydrochloride | FUJIFILM Wako Chemicals | 160-23151 |
| 2% uranyl acetate | Electron Microscopy Sciences | 22400-2 |
| 8% glutaraldehyde | Electron Microscopy Sciences | 16000 |
| 16% paraformaldehyde | Electron Microscopy Sciences | 15700 |
| 4% osmium tetroxide | Electron Microscopy Sciences | 19100 |
| EPON-812 | Electron Microscopy Sciences | 14900 |
| Aclar-film | Nisshin-EM | 453 |
| Soium cacodylate | FUJIFILM Wako Chemicals | 194-04852 |
| DAB(3,3'-Diaminobenzidine Tetrahydrochloride) | FUJIFILM Wako Chemicals | 349-00903 |
| Hydrogen Peroxide | FUJIFILM Wako Chemicals | 081-04215 |
| **Software** | | |
| Fiji/ImageJ | https://imagej.net/software/fiji/ | |
| Volocity | https://www.volocity4d.com | |
| **Other** | | |
| FV3000 | https://www.olympus-lifescience.com/ja/laser-scanning/fv3000/ | |
| SCLIM | Kurokawa et al, 2013 | |

| Reagent/Resource | Reference or Source | Identifier or Catalog Number |
|---|---|---|
| Ultramicrotome EM UC7 | Leica | |
| SEM Regulus8240 | Hitachi | |
| TEM JEM1400 | JEOL | |

## Construction of plasmids for RudLOV and organelle markers

The fluorescent tags described herein, Clover, Ruby, NeonGreen, and Scarlet, are tandem repeats of two identical fluorescent proteins, mClover3, mRuby3 (gift from Michael Lin; (Bajar et al, 2016), and mNeonGreen and mScarlet-I (gift from Erik Dent; (Taylor et al, 2020), respectively. Details of the plasmid vectors used in this study are described in Dataset EV1. Sequence files in the GenBank format are available in Dryad (https://doi.org/10.5061/dryad.jm63xsjhs).

## Stably transformed HeLa cells expressing LOV2 hook—hSec61b::TagBFP2::LOV2

HeLa cells stably expressing GalT::iRFP713 (clone C9; (Fujii et al, 2020a) were transfected with pEBP-Sec61b-mTagBFP2-LOV2 using jetPRIME transfection reagent (Polyplus-transfection, Ill-kirch-Graffenstaden, France). After 24 h, the cells were selected using 2 µg/mL of puromycin (Fujifilm WAKO Chemicals, Osaka, Japan). The clones of stable transformants were established using limited dilutions. A clone showing strong TagBFP2 expression was selected for further experiments (clone G7).

## Transferrin uptake in HeLa cells

After 4 h of treatment with 10 µM nocodazole (Cayman Chemical, Ann Arber, USA), GalT::iRFP713 expressing HeLa cells (Clone C9) were preincubated for 30 min in serum-free medium containing nocodazole. The cells were then incubated with 30 µg/mL of Alexa Fluor 568 conjugated Tfn (Tfn-568; Life Technologies, Carlsbad, CA, USA) for 5 min and chased for 8 min in nocodazole-containing medium. Dynasore (100 µM; Cayman Chemical) was added to the serum-free medium, medium with Tfn, and chasing medium.

## Live imaging of HeLa cells using RudLOV by FV3000

HeLa cells stably expressing GalT::iRFP713 and Sec61b::TagBF-P2::LOV2 (clone G7) were transfected with a DNA plasmid encoding Zdk1-FP-GPI, Zdk1-FP-VSVG, or TNFα-Zdk1-FP using jetPRIME or JetOptimus transfection reagent according to the manufacturer's instructions. At 6–10 h after transfection, the medium was replaced with fresh medium containing 25 µM of biliverdin (Cayman Chemical). The next day, the cells were treated for 4 h with nocodazole. Cycloheximide (100 µg/mL; Cayman Chemical) was administered, and Zdk1-FP-cargo release into the secretory pathway was induced by administration of 0.05% 445 nm laser illumination for 5–10 min using the LSM stimulation mode in FV3000 (Evident, Tokyo, Japan), equipped with a quinta Band

dichroic mirror 89903bs (Chroma Technology Corporation, Bellows Falls, VT, USA). Time lapse 3D-stacks of confocal micrographs were obtained using an FV3000 (except for those shown in Fig. 1F). The 3D-stacks were projected using the maximum intensity with Fiji. The obtained 3D voxel data were displayed as volume-rendered images using the Volocity software. Figure 1F shows single sections of the confocal micrograph for pre-illumination and 15 min post-illumination conditions, obtained using an FV3000. As shown in Fig. 1G, confocal micrographs were obtained, and the signal intensity of the Zdk1-FP-cargo was measured in the Golgi area defined by GalT::iRFP713 using Fiji. Precise cargo localization within the Golgi stacks was measured using line profiles across the Golgi stacks using Fiji, and processed using Plot2 Pro (Michael Wesemann, Berlin, Germany). Co-localization of VSVG and GM130/GalT was measured using custom plugins for Fiji, as described by Papanikou et al (Papanikou et al, 2015). Briefly, for each time point, slices in the Z-stack were concatenated into a montage, regions positive to GM130, GalT, and Cargo were defined, and the Mander's overlap coefficient (MOC) were calculated for each montage.

We administered 100 µM of dynasore, 10 µM of Dyngo4a (Cayman Chemical), 10 µM of dynole34-2, and 100 µM of dynole31-2 (negative control; R&D Systems, Minneapolis, MN, USA) before or after illumination, as indicated in the text.

## Live imaging of HeLa cells using the RUSH system with an FV3000 confocal microscope

HeLa cells stably expressing GalT::iRFP713 were transfected with a DNA plasmid encoding RUSH system bi-cistronic expression plasmids Str::KDEL_SBP::NeonGreen::VSVG using jetPRIME or JetOptimus transfection reagent as per the manufacturer's instructions. At 6–10 h after transfection, the medium was replaced with fresh medium containing 25 µM of biliverdin. The next day, after 4 h of treatment with nocodazole, the release of SBP::Neon-Green::VSVG into the secretory pathway was induced by replacing the medium with 100 µM biotin (Fujifilm WAKO Chemicals) and 100 µg/mL of cycloheximide. Time-lapse confocal micrographs were obtained using an FV3000 microscope. As shown in Fig. 1B, the signal intensity of SBP::Neon Green::VSVG was measured in proximity to the Golgi apparatus using Fiji as defined by GalT::iRFP713.

## Live imaging of HeLa cells using the RUSH system with SCLIM

HeLa cells stably expressing GalT::iRFP713 inoculated on glass-based dishes (Iwaki, Tokyo, Japan) were transfected with a DNA plasmid encoding the RUSH system bi-cistronic expression plasmids Str::KDEL_SBP::Scarlet::GPI using JetOptimus transfection reagent as per the manufacturer's instructions. The cells were cultured for 1 day in phenol red-free medium to reduce background fluorescence. The cells were then treated with 10 µM nocodazole for 4 h to disrupt the microtubules. SBP::EGFP::GPI was released into the secretory pathway by replacing the medium with 100 µM biotin and 100 µg/mL of cycloheximide. To observe the effect of Dyngo4a, the cells were pre-treated with 100 µM of Dyngo4a for approximately 10 min. The medium was replaced with another containing 100 µM biotin, 100 µg/mL cycloheximide, and 100 µM Dyngo4a to

initiate the transport of SBP::EGFP::GPI. Z-stack images were obtained using SCLIM (Kurokawa et al, 2013; Kurokawa et al, 2019; Tojima et al, 2023) and processed via deconvolution with Volocity (Perkin Elmer, Waltham, MA, USA) using the theoretical point-spread function for spinning-disk confocal microscopy. The obtained 3D voxel data were displayed as volume-rendered images using the Volocity software.

## Electron microscopy imaging of GalT::APEX2::EGFP expressing HeLa cells with and without dynasore treatment

Aclar-film (Nisshin-EM, Tokyo, Japan) was cut into pieces, washed with acetone, placed in an μ-Slide 8-well cell culture chamber (ibidi GmbH, Gräfelfing, Germany), and hydrated using culture medium for more than 24 h. HeLa cells that were transfected with CMV-GalT::APEX2::EGFP or CT7-NeonGreen-n::APEX2::eCOP and seeded on the film to grow overnight. After 1 h of incubation with 100 μM of dynasore (Cayman Chemical), the cells were fixed in 0.1 M cacodylate buffer (pH 7.4) with 2% glutaraldehyde (Electron Microscopy Sciences, Hatfield, PA, USA) and 2% paraformaldehyde (Electron Microscopy Sciences) for 1 h on ice. DAB staining was performed as described by Martell et al (Martell et al, 2017) with some previously described modifications (Otsuka et al, 2019). DAB staining was performed by applying fresh DAB solution every 20 min for 1 h at room temperature to cells expressing GalT::APEX2::EGFP and NeonGreen::APEX2::-eCOP. Post-fixation was performed using 2% (w/v) osmium tetroxide (Electron Microscopy Sciences, Hatfield, UAS) for 30 min in chilled cacodylate buffer. The films were rinsed five times (2 min each time) with chilled distilled water, placed overnight in ddH$_2$O containing chilled 2% (w/v) uranyl acetate, dehydrated, and penetrated with EPON-812 resin (Electron Microscopy Sciences) as described previously (Otsuka et al, 2019). Aclar-films with cells in EPON-812 were polymerized into a thin EPON-812 sheet at 100 °C for 20 h. Polymerized EPON-812 sheet-containing cells were cut into 2 mm squares, and the attached Aclar-film was removed and re-polymerized after being placed on the bottom of a pyramidal mold with new EPON-812. The cells were cut with a diamond knife into 70-nm sections and imaged using a JEM1400 transmission electron microscope (JEOL, Tokyo, Japan) operated at 80 kV. Montage images were captured with a CCD camera system (JEOL).

## Serial section scanning electron microscopy observation of GalT::APEX2::EGFP-expressing HeLa cells

Serial section scanning electron microscopy was performed using a high-resolution field-emission scanning electron microscope and back-scattered electron detector. Serial ultrathin sections (thickness of 50 nm) were cut using a diamond knife (Diatome 45°) on an ultramicrotome (EM UC7, Leica Microsystems, Wetzlar, Germany) and placed on silicon wafers (10 × 22 mm). The sections were stained with 0.4% uranyl acetate for 10 min and lead stain solution (Sigma-Aldrich, St. Louis, MO, USA) for 2 min, and then coated with osmium tetroxide using an osmium coater (HPC-1SW, Vacuum Device Inc., Mito, Japan). Serial sections were observed using a field-emission scanning electron microscope (Regulus8240; Hitachi High-Tech, Tokyo, Japan) equipped with auto-capture for array tomography and a low-angle back-scattered electron detector at an accelerating voltage of 2 kV.

## Data availability

Data are available from Dryad: https://doi.org/10.5061/dryad.jm63xsjhs and BioStudies database: https://www.ebi.ac.uk/biostudies/bioimages/studies/S-BIAD1429 (Sarkans et al, 2018).

The source data of this paper are collected in the following database record: biostudies:S-SCDT-10_1038-S44319-024-00342-z.

## Peer review information

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

## Acknowledgements

This work was supported by the Japan Society for the Promotion of Science [JSPS] (KAKENHI grant no. 22H02617 to AKS; KAKENHI grant no. 19K06566 to TS; KAKENHI grant no. 22K06213 to TT[3]; KAKENHI grant no. JP22H04926 to KT, KAKENHI grant no. 18H05275 and 23H00382 to AN), Japan Science and Technology Agency [JST] (PRESTO grant no. 25-J-J4215 and CREST grant no. JPMJCR22E2 to AKS; CREST grant no. JPMJCR21E3 to TT[3]; SPRING grant no. JPMJSP2132 to TT[1]), Takeda Science Foundation, Ohsumi Frontier Science Foundation (Grant Number JP22H04926) to AKS. We thank Rumi Sato and Takaharu Okada (RIKEN IMS) for offering the field emission scanning electron microscope equipped with auto-capture for array tomography. We thank Natural Science Center for Basic Research and Development at Horoshima University for use of transmission electron microscopy JEM1400 and confocal microscopy FV3000. We would like to thank Editage (http://www.editage.com) for English language editing of this manuscript.

## Author contributions

**Tatsuya Tago**: Funding acquisition; Investigation; Writing—review and editing. **Takumi Ogawa**: Investigation; Writing—review and editing. **Yumi Goto**: Investigation; Writing—review and editing. **Kiminori Toyooka**: Investigation; Writing—review and editing. **Takuro Tojima**: Supervision; Writing—review and editing. **Akihiko Nakano**: Supervision; Writing—review and editing. **Takunori Satoh**: Supervision; Funding acquisition; Methodology; Writing—original draft; Project administration. **Akiko K Satoh**: Supervision; Funding acquisition; Methodology; Writing—original draft; Project administration.

Source data underlying figure panels in this paper may have individual authorship assigned. Where available, figure panel/source data authorship is listed in the following database record: biostudies:S-SCDT-10_1038-S44319-024-00342-z.

## Disclosure and competing interests statement

All authors have read and approved this work and declare that they have no financial conflicts of interest.

# Expanded View Figures

**Figure EV1.  Delay of cargo movement in cells with high hook/cargo expression in the RUSH system.**

(**A**) Double-colored images with SBP::NeonGreen::VSVG (green) and GalT::iRFP713 (magenta) of Fig. 1A. (**B**) Dot plot of the relative expression level of the cargo in each cell, categorized by different time-to peak of SBP::NeonGreen::VSVG after biotin administration in the Golgi apparatus in the RUSH system. (**C**) Localization of SBP::NeonGreen::VSVG against Golgi stack at 0 min (left) and 10 min (right) after the first observation (0 min) without transport-trigger. Golgi markers (TagBFP2::GM130, GalT::iRFP713) and ERES marker (Scarlet::Sec13) are shown in the indicated colors. Plots show signal intensities from the image on the upper ERES/Golgi unit. Signal intensity was measured along the arrow (representing 1.5 μm). (**D**) Double-colored images with Zdk1::Clover::VSVG (green) and GalT::iRFP713 (magenta) of Fig. 1F. (**E, F**) Localization of Zdk1::Clover::GPI before (left) and at 10, 15, 30, 40, 50, and 70 min after onset of illumination at 445 nm using the RudLOV system in untreated (**E**) and nocodazole-treated cells (**F**). Inset in (**F**) shows a magnified image of Zdk1::Clover::GPI. The cargo is shown in green and GalT::iRFP713 in magenta. (**G**) Localizations of TNFα::Zdk1::Clover before (left) and at 10, 15, 20, 30, 50, and 60 min after onset of illumination with 445 nm using the RudLOV system. The cargo is shown in green, Ruby::GM130 in red, and GalT::iRFP713 in blue. (**H**) Localizations of TNFα::Zdk1::Clover before (left) and at 10, 20, 30, 50, and 70 min after onset of illumination with 445 nm using the RudLOV system in nocodazole-treated cells. Inset shows the magnified image of TNFα::Zdk1::Clover. The cargo is shown in green, Ruby::GM130 in red, and GalT::iRFP713 in blue. Data in (**A**) and (**C–H**) are representative of more than three replicates. Scale bars: 20 μm (**A, D**), 5 μm (**E–H**), and 2 μm (insets in **F, H**).

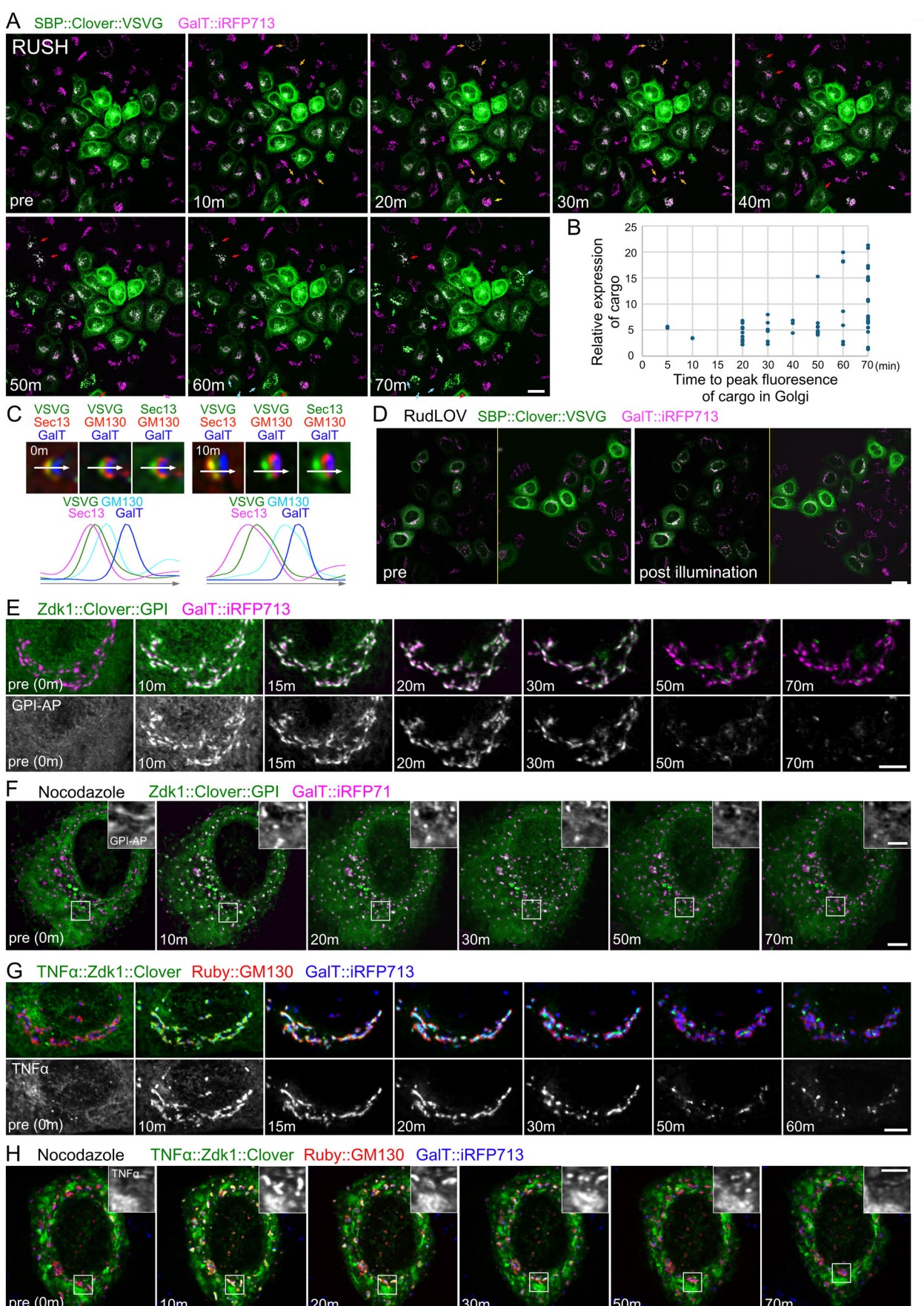

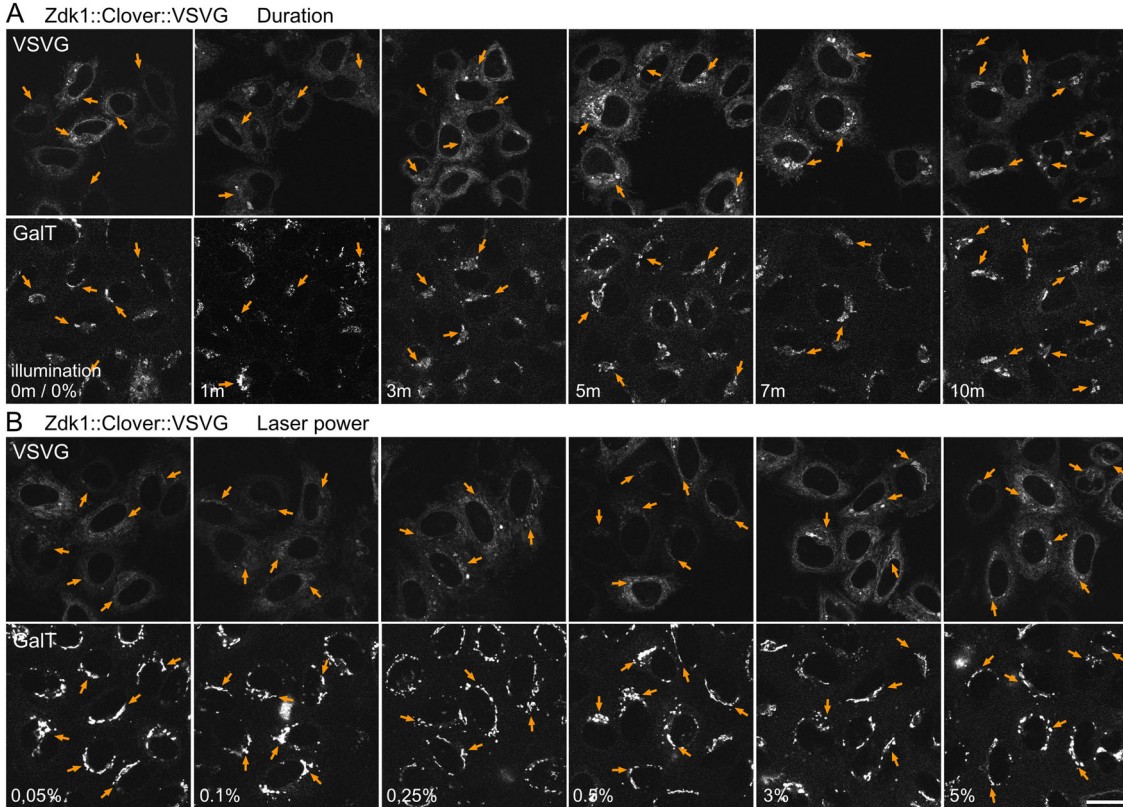

**Figure EV2. RudLOV enables quantitative control of cargo release.**

(A) Localization of Zdk1::Clover::VSVG (upper panel) or GalT::iRFP713 (lower panel) 10 min after the start of 0, 1, 3, 5, and 10 min illumination at 445 nm with 3% laser power using the RudLOV system. Arrows indicate Golgi stacks. (B) Localization of Zdk1::Clover::VSVG (upper panel) or GalT::iRFP713 (lower panel) after 5 min illumination at 445 nm with 0.05%, 0.1%, 0.25%, 0.5%, 3%, and 5% laser power using the RudLOV system. Arrows indicate Golgi stacks. Data in (A) and (B) are representative of more than three replicates. Scale bars: 20 μm (A, B).

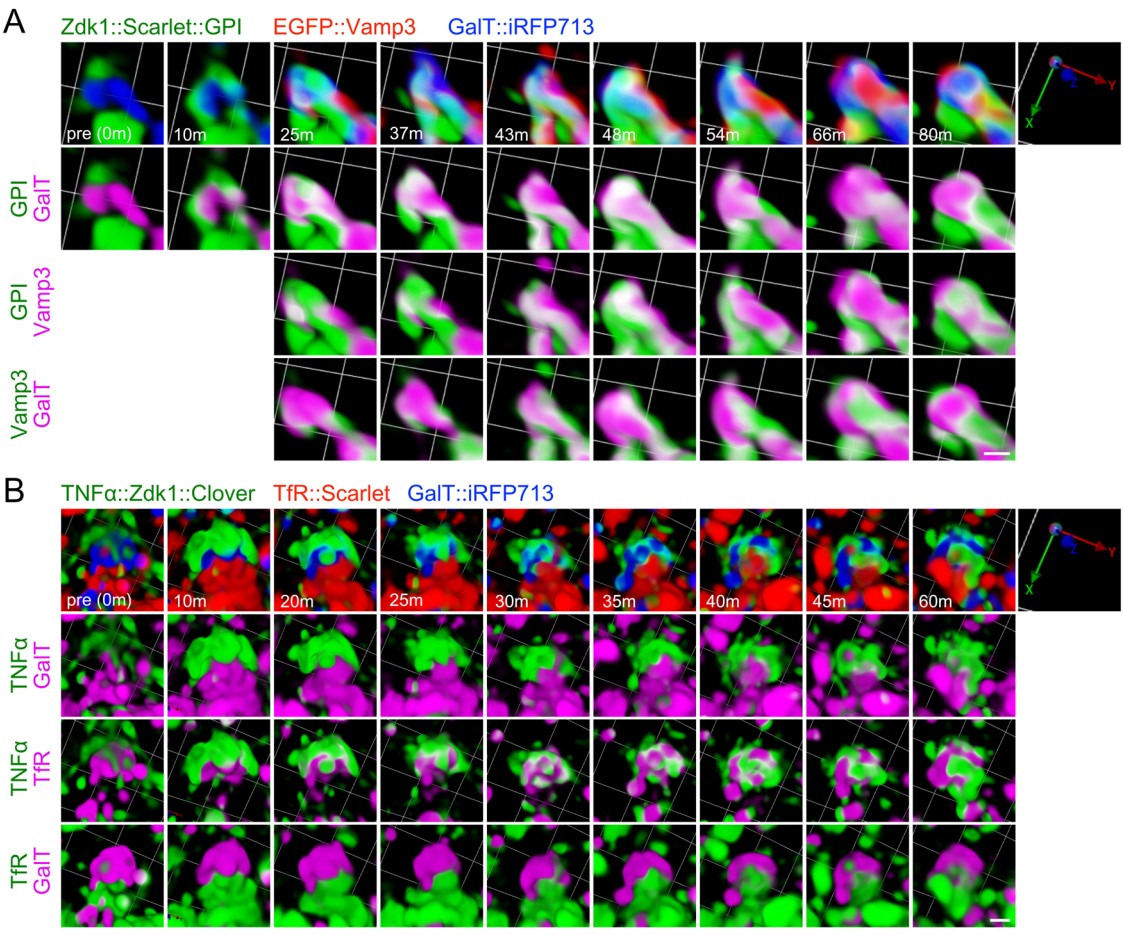

**Figure EV3.  TNFα movements within Golgi/RE unit observed by RudLOV.**

(A) Localization of the cargo Zdk1::Scarlet::GPI before (left) and after illumination in a single Golgi/RE unit in nocodazole-treated cells. The time after illumination is shown in the bottom-left corner. The cargo is shown in green, *trans*-Golgi marker GalT::iRFP713 in blue, and RE marker EGFP::Vamp3 in red (upper panel). Double-colored images separated from triple-colored images (lower panels). (B) Localization of the cargo TNFα::Zdk1::Clover before (left) and after illumination in a single Golgi/RE unit in nocodazole-treated cells. The time after illumination is shown in the bottom-left corner. The cargo is shown in green, *trans*-Golgi marker GalT::iRFP713 in blue, and RE marker TfR::Scarlet in red (upper panel). Double-colored images separated from triple-colored images (lower panels). Data in (A) and (B) are representative of more than three replicates. Scale bars: 1 μm (A, B).

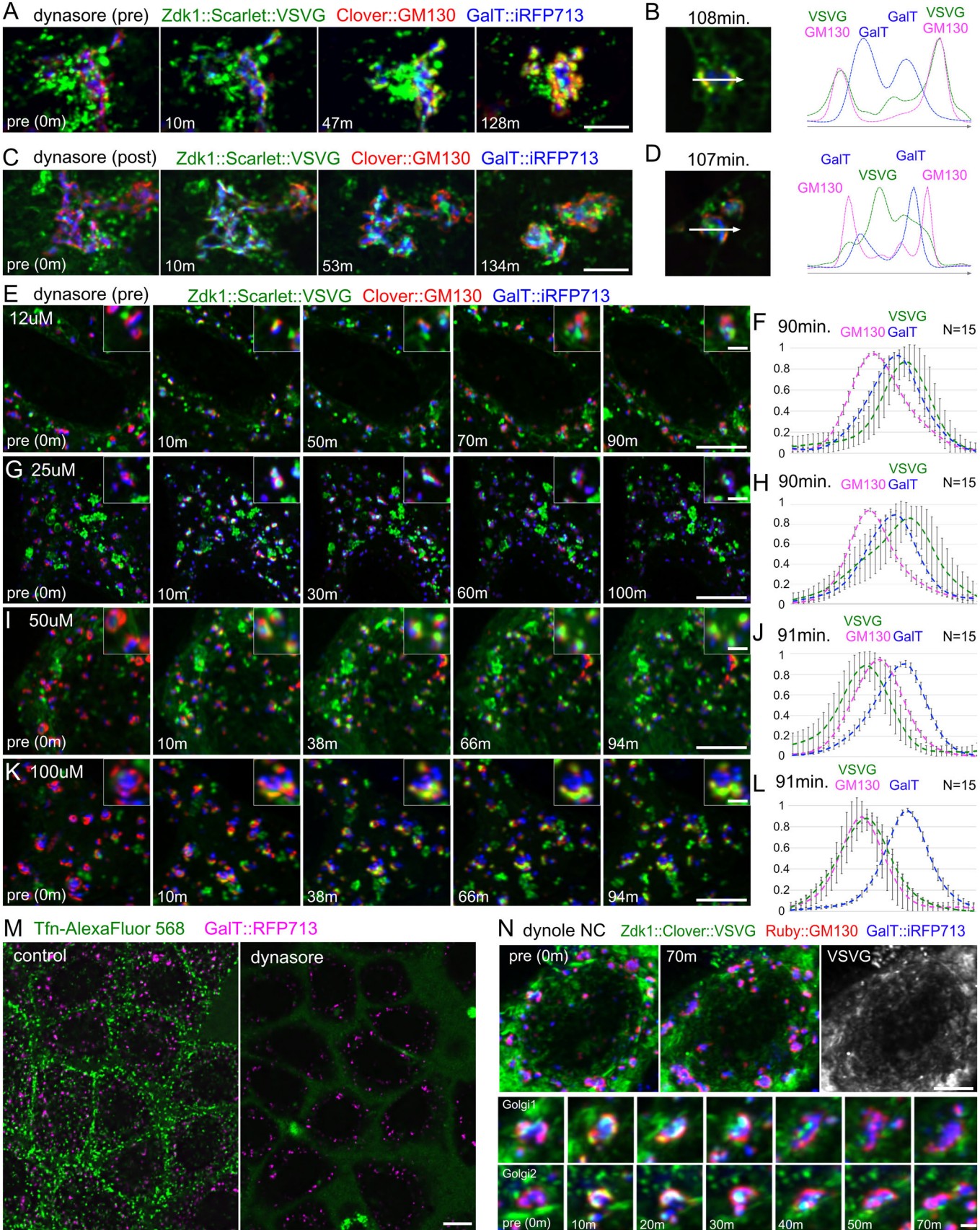

**Figure EV4.  Dynasore inhibits cargo transport at the *cis*- or *trans*-side of Golgi stacks in nocodazole-untreated cells.**

(A) Localization of Zdk1::Scarlet::VSVG before illumination and at 10, 47, and 128 min after onset of illumination in cells pre-treated with dynasore. The cargo is in green, *cis*-Golgi marker Clover::GM130 in red, and *trans*-Golgi marker GalT::iRFP713 in blue. (B) Plots showing signal intensities from the image of the left Golgi/RE unit as 108 min after onset of illumination in the cell pre-administered with dynasore. Signal intensity was measured along the arrow (representing 3 µm). (C) Localization of Zdk1::Scarlet::VSVG before illumination and at 10, 53, and 134 min after onset of illumination in cells that were post-administrated dynasore 4 min after illumination. The cargo is in green, *cis*-Golgi marker Clover::GM130 in red, and *trans*-Golgi marker GalT::iRFP713 in blue. (D) Signal intensities from the image of the left Golgi/RE unit 107 min after onset of illumination in a dynasore pre-administered cell. Signal intensity was measured along the arrow (representing 3 µm). (E–L) Localization of Zdk1::Scarlet::VSVG before illumination (left) and after illumination in cells pre-treated with dynasore at 100, 50, 25, and 12 µM (E, G, I, K). Insets in (E), (G), (I), and (K) show the magnified image of a single Golgi stack. Dynasore was administrated 1 min before illumination. Zdk1::Scarlet::VSVG is shown in green, *cis*-Golgi marker Clover::GM130 in red, and *trans*-Golgi marker GalT::iRFP713 in blue. Plots show the normalized means of 15 line profiles of Zdk1::Scarlet::VSVG, Clover::GM130 and GalT::iRFP713 across the Golgi stack at 90 min (F, H) and 91 min (J, L) after the start of illumination. The 15 line profiles were obtained from five Golgi stacks per cell, using three different cells. Error bars are presented as mean ± SD. (M) Uptake of Tfn in untreated and dynasore-treated cells 8 min after incubation with 30 µg/ml of Alexa Fluor 568-conjugated Tfn. (N) Localization of Zdk1::Clover::VSVG before illumination (upper left) and 70 min after illumination (upper middle and right) in cells pre-treated with dynole 31-2 (negative control: NC). Time-course of Zdk1::Clover::VSVG localization before and after illumination in a dynole NC-treated single Golgi stack (bottom). Zdk1::Clover::VSVG is shown in green, *cis*-Golgi marker Ruby::GM130 in red, and *trans*-Golgi marker GalT::iRFP713 in blue. Data in (A), (C), (E), (G), (I), (K), (M) and (N) are representative of more than three replicates. Scale bars: 5 µm (A, C, E, G, I, K), 1 µm (insets in E, G, I, K), 10 µm (M), 5 µm (upper panel in N), and 1 µm (lower panel in N).

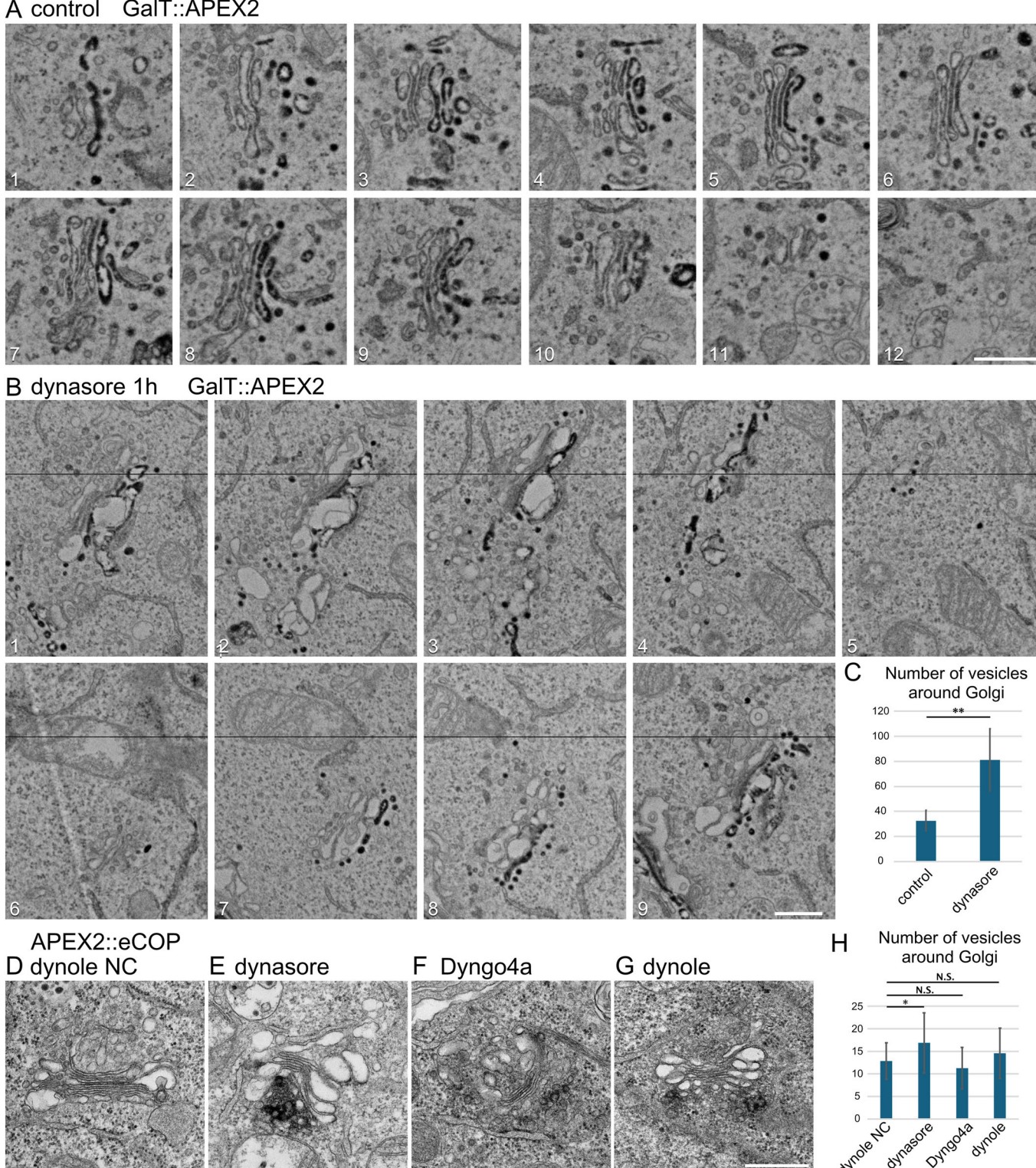

**Figure EV5. Accumulation of vesicles at the *cis*-side of Golgi stacks after administration of dynasore.**

(A, B) Scanning electron micrographs of serial sections of a Golgi stack at 150 nm-intervals in the cell after 1 h of incubation with (B) or without 100 µM dynasore (A). GalT::APEX2 visualized trans-Golgi cisternae and vesicles. (C) Number of vesicles accumulated near Golgi stacks in the cell after 1 h of incubation with or without 100 µM dynasore. Error bars indicate the standard deviation of five Golgi stacks. Significance according to two-tailed unpaired Student's t-test (Welch's t-test), where $**p < 0.01$. Error bars are presented as mean ± SD. Exact P values (to 4 decimal points) for (C) 0.0072. (D–G) Transmission electron micrographs of Golgi stacks with APEX2::eCOP showing COPI budding profiles and vesicles in the cell after 1 h of incubation with dynole NC (D), dynasore (E), Dyngo4a (F), and dynole (G). (H) Number of vesicles accumulated near Golgi stacks in the cross sections of cells after 1 h of incubation with dynasore, dynole NC, dynole, or Dyngo4a. Error bars indicate the standard deviation of $N > 13$ sections of Golgi stacks. Significance according to two-tailed unpaired Student's t-test (Welch's t-test), where $*p < 0.05$. Error bars are presented as mean ± SD. Exact P values (to 4 decimal points) for (H) dynoleNC vs dynasore 0.0444, dynoleNC vs Dyngo4a 0.3589 and dynoleNC vs dynole 0.3452. Data in (A), (B) and (D–G) are representative of more than three replicates. Scale bars: 500 nm (A, B, D–G).

