## [Peer Review File · EMBO Reports]

RudLOV-a new optically synchronized cargo transport method reveals unexpected effect of dynasore

Tatsuya Tago, Takumi Ogawa, Yumi Goto, Kiminori Toyooka, Takuro Tojima, Akihiko Nakano, Takunori Satoh, and Akiko Satoh

Corresponding author(s): Akiko Satoh (aksatoh@hiroshima-u.ac.jp) , Takunori Satoh (tsatoh3@hiroshima-u.ac.jp)

Review Timeline:

Submission Date:	8th Nov 23
Editorial Decision:	10th Jan 24
Appeal Received:	31st Jul 24
Editorial Decision:	13th Sep 24
Revision Received:	24th Sep 24
Accepted:	13th Nov 24

Editor: Deniz Senyilmaz Tiebe

Transaction Report:

Dear Dr. Satoh,

Thank you for submitting your manuscript to EMBO Reports. We have now received three referee reports, which are included below.

Please accept my apologies for this unusual delay in getting back to you. It took longer than anticipated to receive the full set of referee reports.

The referees express interest in the presented method for synchronizing cargo release. However, referees also raise significant concerns. In particular,

- The comparison between RudLOV and RUSH methods needs better experimental support (referee #2 paragraphs 2 and 3, referee #3 point 1).
- As it stands, the advantage of RudLOV over existing methods is not clear (referees #2 and 3).

These are important points that need to be addressed for publication here. Given such input from these recognized experts who are also experienced referees, and the amount of work required to address these concerns, we cannot publish your manuscript in its current form.

However, in case you feel that you can address the referee concerns in a timely and thorough manner, and can obtain data that would considerably strengthen the study as in the referee reports, we would be happy to consider a revised manuscript (along with a point-by-point response to the referee concerns). Please note that if you were to send a new manuscript this would be assessed again with respect to the literature and the novelty of your findings at the time of resubmission and in case of a positive editorial evaluation, the manuscript would be sent back to the original referees. I would like to emphasize that we will be reluctant to approach the referees again in the absence of major revisions, and we need strong support from the referees to consider publication here.

That being said, I have discussed your manuscript and the accompanying reviews with the Executive Editor our not-for-profit open-access sister journal, Life Science Alliance (LSA), Eric Sawey, who is interested in these findings, and would like to invite further consideration of this manuscript at LSA pending revisions with a narrower scope:

- Address Reviewer 1's comments.
- Address Reviewer 2's comments outlining areas that need statistical analysis, and the need for a positive control for the dynasore experiment. The comments regarding the comparison to RUSH can be addressed as outlined, or by tempering those conclusions.
- Address Reviewer 3's comments, except for Main comment #2.

We understand that such a revision might need to be re-reviewed, in which case, Dr. Sawey will walk the Reviewers through our transfer process.

We encourage you to use the link below to transfer your manuscript to LSA. You do not need to revise the manuscript before transferring it to LSA. Once you transfer, Dr. Sawey will email you an invitation to revise and resubmit, listing the same revision requests as mentioned above. Please feel free to reach out at e.sawey@life-science-alliance.org if you have any questions about the LSA journal, the transfer process or the revisions requested.

Thank you for the opportunity to consider this manuscript.

Kind regards,

Deniz Senyilmaz Tiebe

Deniz Senyilmaz Tiebe, PhD
Editor
EMBO Reports

Referee #1:

This is an exciting manuscript in which the authors develop a new method for synchronizing cargo release from an organelle. The method uses LOV2 and therefore cargo can be released with high spatial and temporal control by using focused light. The method also allows for dose-dependent release by using different amounts of light. In principle cargo could be stored and then

released from any organelle, and as a proof-of-principle the authors use the ER for this study, which enables them to track cargos as they exit the ER and then continue throughout the secretory pathway.

The authors validate this new tool using several different cargos, and they demonstrate the spatial control possibilities by stimulating cargo release in just a single cell within a field of cells.

They track "small amounts" of released cargos and find that different cargos take different routes during secretion, for example, GPI-AP transits through the RE compartment but VSVG does not. These experiments also provide further evidence that Golgi maturation occurs in cultured human cells.

The authors then apply this tool to study the effects of dynamin protein addition by the inhibitor dynasore. Dynasore is a known endocytosis inhibitor but its effects on Golgi trafficking are less established. The authors make the surprising observation that multiple cargos become stalled in the cis-Golgi after dynasore treatment.

They then use EM analysis to determine that dynasore induces accumulation of many vesicles at the cis-face of the Golgi, suggesting dynasore inhibits tethering/fusion of vesicles with the cis-Golgi.

Using a staged experiment, the authors find that addition of dynasore after cargo release from the ER instead blocks the cargo at the TGN. This was expected based on the reported role of dynamin-2 at the TGN.

In contrast to dynasore, the authors find that another dynamin inhibitor, Dyngo4a inhibits cargo transport at the TGN but not at the cis-Golgi. They therefore conclude that dynasore may have an additional protein target that is important for cis-Golgi trafficking.

Overall I think this is a nice study reporting a new tool that will be very useful for studying membrane trafficking and organelle homeostasis. The authors do a good job of validating the tool and highlighting possible uses. They also make the surprising finding that dynasore inhibits trafficking at a location that was not previously appreciated, suggesting an unknown target that can be identified in future studies. I have only one suggested experiment needed to provide support for one of the claims regarding the RudLOV tool, listed as a major point below.

Major point:

1. The authors draw attention to the ability of RudLOV to be used for "quantitative control" of cargo release. As an example, in fig 2 they claim to use low levels of light to release "small amounts" of cargo. However, the reader is left wondering just how tunable this 'quantitative control' actually is. Therefore, if the authors want to make claims about quantitative control, ideally they would present an experiment demonstrating how different amounts of cargos can be released with different doses of light.

Minor points:

2. In fig. 2, three cargos are tracked but different sets of Golgi/RE markers are followed in each. While a direct comparison is made between VSVG and GPI-AP, the analysis of TNFalpha is different because different markers (VAMP3 and Rab6) are used. Therefore, although it is clear that VSVG and GPI-AP follow different routes from each other, it is hard to relate the route followed by TNFalpha to the other two cargos. I suggest the authors either provide additional direct comparisons or else tone down the interpretations of TNFalpha following a different route.

3. For the experiments shown in Figure 3C-3H, how long were the cells pre-treated with dynasore prior to illumination? This information is very important and should be clearly presented in the Figure, Figure Legend, and Results text.

4. If the dynasore pre-treatment time is long (i.e. multiple-hours) it is possible that they are observing an indirect effect of blocking traffic at another compartment. Depending on the length of dynasore treatment, the authors should comment on this possibility.

5. On line 246, I think 'Figure 3F' should be 'Figure 4F'.

Referee #2:

Tago et al have presented a novel tool for the synchronous trafficking of cargo through the secretory and endolysosomal system. Their system, named RudLOV, takes advantage of the light-dependent release of the LOV2 C-terminal helix Jx and Zdk1. They tether Zdk1 to their cargo and LOV2 to sec61 (an ER localised membrane protein), which allows retention when not in the presence of 445nm light. They argue that their system demonstrates better kinetics and spatial specificity than the commonly used RUSH system. In addition, they argue it is more economically efficient and less toxic than the newer zapERtrap system. Using their new system, they demonstrate cargo trafficking through the Golgi apparatus and use Golgi ministacks to

observe trafficking through the Golgi/RE. They then tested the effects of the Dynamin-2 inhibitor Dynasore and unexpectedly saw an earlier block than previously observed. Interestingly, they show that this is not due to the effects of blocking dynamin-2 as other inhibitors do not have the same effect, they thus conclude that there is an off target effect for some cis-Golgi uncoating, tethering, or fusion machinery. On the whole the manuscript is clear and well written and the core of the manuscript the RudLOV system a valuable contribution to the field. The specific effect of the off-target by Dynasore is also interesting and novel; however, not as important as the system itself. I have some comments concerning some of the comparative measures, statistics and controls that I hope will improve this useful contribution.

In Figure 1, the kinetics of RUSH and RudLOV are compared. The authors conclude that the RUSH is less efficient. This may well be true; however, it is hard to draw that comparison as the comparison is not fair due to the use of different hooks. I think this is rather important as if this system is to be compared; it would be useful to do a fair comparison of the two systems. It is also absolutely imperative to know what the time is between "pre" and "illumination"; if this is 10 mins, then the system is comparable to RUSH. Finally, it is also essential to know the expression of the cargo. It is well known that cells overexpressing cargo RUSH with slower kinetics, and the cells in Panel D appear, to my eyes, to be overexpressing the cargo. This can be quantified with total GFP. If all these things are accounted for, this will be a much more useful comparison.

In addition, from my reading of the methods, the RudLOV system is stably integrated in the cells, while the RUSH system is transiently transfected. It is well known that transient transfections yield much higher expression, and also, the transfection itself affects trafficking- I think the authors need to be much more careful with the comparison or more careful with the conclusion. Really, this is the key to the manuscript, as at least being comparable to RUSH is enough to make this interesting.

It is not clear here how many independent repeats of this and all experiments have been performed, this is absolutely crucial for these sorts of observations.

In addition, here and throughout the manuscript, the use of statistics is either not present or improper. As I said, this is the most important figure for the manuscript, in my opinion, so the findings need to be well-validated.

In Figures 2, 3, and 4, cargo localisation is quantified through line scans. This has two issues: 1) the drawing of the line can affect the observation, and 2) it lacks statistical output. I think that to make these interpretable findings; it would be very useful to have proper statistical analysis here from multiple cells across multiple experiments.

The authors conclude that dynasore has an unexpected off-target effect. However, the concentration was not explored, and no positive control was used for the dynasore, thus, perhaps there is a problem with that batch of dynasore or that stock that causes the observed phenomenon.

I find the speculation of the actual molecular target of the off-target effect lacking evidence, and although I leave it to the authors, without evidence, it is perhaps better not to make a claim on the target.

The statistical comparison in Figure 4F is not valid. It is a student's T-test across a time series, which is not a proper use of that test. I advise the authors to get advice on the proper statistical test to perform here and throughout.

Referee #3:

In this study, Tago et al describe a new system of controlled cargo release within the secretory pathway and use it to demonstrate that treatment of cells with the dynamin inhibitor dynasore impacts two steps in the secretory pathway - cis/medial Golgi maturation and TGN exit. They then show that only the latter is affected by other dynamin inhibitors and thus dynasore is likely targeting a different protein to regulate cisternal maturation. Unfortunately, this target remains unidentified but this finding serves as a warning to dynasore users.

The behaviour of their new RudLOV2 system for controlled cargo release seems robust and has superior spatial resolution over the current predominant alternative, RUSH. It does however appear to have limitations as the illumination period is restrictive for imaging early ER-Golgi dynamics and it cannot be used with certain experimental set ups - the authors themselves revert to using RUSH for some experiments. Nonetheless it may be of use to some in the membrane trafficking field looking to diversify their toolset.

It is worth noting that other photoactivatable transport systems have already been reported, for example UVR8 from the Kennedy group. It is not clear from this manuscript what advantage RudLOV2 has over these comparable systems and this should be discussed in the manuscript.

Whilst the majority of the data is well presented, there are two main areas I believe require improvement to strengthen the premise and conclusions of this study:

1) With respect to the RUSH experiments, I do not think the data, as currently presented, support the conclusion that higher expression levels result in delayed cargo export as reported in line 107-108. The quantification provided only shows time of enrichment in the Golgi, without any correlation to expression levels. Plotting time against signal intensity for example would better represent this conclusion.

I am also concerned that the VSVG-RUSH construct appears to be leaky - the VSVG signal looks to be in the Golgi region before treatment in most cells, and so it is unclear which cells they have chosen to quantify and how this will impact the dynamics. There are other RUSH cargoes which are less leaky that the authors could use - and they seem to have TNFa already - or they could increase the ratio of hook:cargo by expressing another hook vector or changing the promoters. Alternatively, as this data is just to set up the premise of this study, there is a lot of literature reporting the dynamics of different cargoes which could be cited to back up this data set.

Finally, according to the materials and methods section, GalT:iRFP713 was co-transfected with these cells. Inclusion of this marker in the figure would help the reader relate cargo location and Golgi enrichment.

2) In lines 222-225 the authors suggest that dynasore 'inhibits the uncoating, tethering or fusion of COPI vesicles' on the basis that there are eCOP-APEX2 labelled vesicles in micrographs of cells treated with dynasore. Whilst this is a good hypothesis that fits with their model, this evidence is not compelling, mainly due to the lack of controls. eCOP-APEX2 would be expected to label vesicles in untreated cells and so it is not remarkable that they are seen here. The authors need untreated control samples and some quantification to show there is an unusual build-up of COPI vesicles in the dynasore treated cells to support their conclusions. It might also be possible to quantify COPI punctae by immunofluorescence depending on how tightly they pack to the peri-Golgi region.

Evidence of vesicle accumulation is also important to the authors model that dynasore is inhibiting a protein required for COPI-dependent cis/medial Golgi maturation, whereas other dynamin inhibitors do not. It would therefore also be worth looking at dyngo4a or dynole 34-2 treated cells by EM too to see whether this also generates a build-up of vesicles. If they don't, this is the step specifically affected by dynasore in the early secretory pathway, and if they do, it is the result of dynamin inhibition at the TGN and not the dynasore-specific step.

Minor points:

1) The authors do not define what LOV2 is. This should be included in the introduction

2) I would suggest that Figure 1 be rearranged so that the data appears in the same order as mentioned in the text - the RUSH data splits up the RudLOV story at the moment.

3) Figure 1G is not referred to in the results/discussion.

4) Figure 1G-I As currently displayed, it is a little confusing to call the first time point after illumination 0min - it looks like translocation is instantaneous. Perhaps this could be labelled differently, have a bar indicating light and dark, or at least in the figure legend state the length of illumination time.

5) Figure 2D-J To better follow the arguments made in the text about the journey of each protein, linescans at multiple relevant timepoints would be more helpful than just the end point. The authors should also be aware that the images in these figures will be uninterpretable to readers with red-green colour blindness and should consider using different colour combinations eg magenta not red.

6) Figure 4G-K are not referenced in the text.

7) The authors do not indicate how many times experiments were repeated as so it is difficult to assess how robust their findings are. This should be included in the figure legends, especially for those which were quantified and statistically analysed.

** As a service to authors, EMBO Press provides authors with the ability to transfer a manuscript that one journal cannot offer to publish to another journal, without the author having to upload the manuscript data again. To transfer your manuscript to another EMBO Press journal using this service, please click on Link Not Available

July 31, 2024

Deniz Senyilmaz Tiebe, Scientific Editor

EMBO reports

Dear Editor,

We are pleased to resubmit our manuscript titled “RudLOV—a new optically synchronized cargo transport method reveals unexpected effect of dynasore” for publication in *EMBO reports* as a report. The paper was co-authored by Tatsuya Tago, Takumi Ogawa, Yumi Goto, Kiminori Toyooka, Takuro Tojima, Akihiko Nakano, and Takunori Satoh. The original journal-supplied reference number for this submission is EMBOR-2023-58431V1.

In this study, we report a new light-induced cargo release method named retention using the dark state of LOV2 (RudLOV). RudLOV has three exceptional control abilities: spatial, temporal, and quantitative control of cargo release. Using RudLOV, we elucidated unexpected inhibition of the well-known dynamin inhibitor dynasore on the cisternal maturation of the early Golgi (*cis/medial to trans*) but not the late Golgi (*trans* to TGN).

In our first submission, the reviewers express interest in this method, but also raise significant concerns. In particular, the reviewers claim that "the comparison between RudLOV and RUSH methods needs better experimental support". Since some of the reviewers' comments seem to be based on the reviewers' misunderstanding of our experiments, we explained them in our “Response to reviewers comments”. We were happy to include new experiments in this revised manuscript, which proved the quantitative control ability of RudLOV for cargo release: stronger illumination releases more cargoes. Although we did not perform the comparison between RudLOV and RUSH system in quantitative and spatial control abilities of cargo release, because they are impossible for RUSH system, we believe the advantage of RudLOV is obvious. We have attempted to address the helpful comments of the reviewers and believe that the manuscript is now much improved.

We confirm that neither the manuscript nor any parts of its content are currently under consideration or published in another journal. All authors have approved the manuscript and agree with its submission to *EMBO Reports*. We have read and

understood your journal's policies, and we believe that neither the manuscript nor the study violates any of these. There are no conflicts of interest to declare.

Thank you for your consideration. We look forward to hearing from you.

Sincerely,

Akiko K. Satoh

Program of Life and Environmental Sciences,

Graduate School of Integrated Sciences for Life, Hiroshima University

1-7-1, Kagamiyama, Higashi-hiroshima, Hiroshima 739-8521, Japan

Tel: +81-52-424-6507

E-mail: aksatoh@hiroshima-u.ac.jp

Referee #1:

This is an exciting manuscript in which the authors develop a new method for synchronizing cargo release from an organelle. The method uses LOV2 and therefore cargo can be released with high spatial and temporal control by using focused light. The method also allows for dose-dependent release by using different amounts of light. In principle cargo could be stored and then released from any organelle, and as a proof-of-principle the authors use the ER for this study, which enables them to track cargos as they exit the ER and then continue throughout the secretory pathway.

The authors validate this new tool using several different cargos, and they demonstrate the spatial control possibilities by stimulating cargo release in just a single cell within a field of cells.

They track "small amounts" of released cargos and find that different cargos take different routes during secretion, for example, GPI-AP transits through the RE compartment but VSVG does not. These experiments also provide further evidence that Golgi maturation occurs in cultured human cells.

The authors then apply this tool to study the effects of dynamin protein addition by the inhibitor dynasore. Dynasore is a known endocytosis inhibitor but its effects on Golgi trafficking are less established. The authors make the surprising observation that multiple cargos become stalled in the cis-Golgi after dynasore treatment.

They then use EM analysis to determine that dynasore induces accumulation of many vesicles at the cis-face of the Golgi, suggesting dynasore inhibits tethering/fusion of vesicles with the cis-Golgi.

Using a staged experiment, the authors find that addition of dynasore after cargo release from the ER instead blocks the cargo at the TGN. This was expected based on the reported role of dynamin-2 at the TGN.

In contrast to dynasore, the authors find that another dynamin inhibitor, Dyngo4a inhibits cargo transport at the TGN but not at the cis-Golgi. They therefore conclude that dynasore may have an additional protein target that is important for cis-Golgi trafficking.

Overall I think this is a nice study reporting a new tool that will be very useful for studying membrane

trafficking and organelle homeostasis. The authors do a good job of validating the tool and highlighting possible uses. They also make the surprising finding that dynasore inhibits trafficking at a location that was not previously appreciated, suggesting an unknown target that can be identified in future studies. I have only one suggested experiment needed to provide support for one of the claims regarding the RudLOV tool, listed as a major point below.

Major point:

1. The authors draw attention to the ability of RudLOV to be used for "quantitative control" of cargo release. As an example, in fig 2 they claim to use low levels of light to release "small amounts" of cargo. However, the reader is left wondering just how tunable this 'quantitative control' actually is. Therefore, if the authors want to make claims about quantitative control, ideally they would present an experiment demonstrating how different amounts of cargos can be released with different doses of light.

We performed two series of experiments to make it clear. One is the investigation of the released cargo amount by the different duration of illumination with the same laser power. The other is the investigation of the released cargo amount by the different laser power with the same illumination time. Obtained graphs are presented as Figure 2D and the original data are shown in Figure EV2. These results show the good 'quantitative control' of RudLOV.

We also added the following sentences in result and discussion.

"Next, we investigated the ability of RudLOV to control the amount of cargo released. We estimated the relative amount of photo-released cargo from the peak intensity of cargo fluorescence in Golgi stacks. Illumination intensity was controlled by the duration or laser power. Stronger illumination resulted in higher peak intensity of cargo fluorescence in the Golgi stacks (Figure 2D, Figure EV2)."

Minor points:

2. In fig. 2, three cargos are tracked but different sets of Golgi/RE markers are followed in each. While a direct comparison is made between VSVG and GPI-AP, the analysis of TNFalpha is different because different markers (VAMP3 and Rab6) are used. Therefore, although it is clear that VSVG and GPI-AP follow different routes from each other, it is hard to relate the route followed by TNFalpha to the other two cargos. I suggest the authors either provide additional direct comparisons or else tone down the interpretations of TNFalpha following a different route.

We added the live imaging observations of GPI with Vamp3 and TNFalpha with TfR in Figure EV3.

3. For the experiments shown in Figure 3C-3H, how long were the cells pre-treated with dynasore prior to illumination? This information is very important and should be clearly presented in the Figure, Figure Legend, and Results text.

In our experiments dynasore was administered 1 min before illumination

We add the following phrase in Results and Discussion,

“When dynasore was administered 1 min before cargo release by illumination,”

We also presented 1 min on Figure and Figure legends.

4. If the dynasore pre-treatment time is long (i.e. multiple-hours) it is possible that they are observing an indirect effect of blocking traffic at another compartment. Depending on the length of dynasore treatment, the authors should comment on this possibility.

As we administered dynasore 1 min before cargo release by illumination, reviewer concern: indirect effect of blocking traffic, is not the case.

5. On line 246, I think 'Figure 3F' should be 'Figure 4F'.

We fixed.

Referee #2:

Tago et al have presented a novel tool for the synchronous trafficking of cargo through the secretory and endolysosomal system. Their system, named RudLOV, takes advantage of the light-dependent release of the LOV2 C-terminal helix J α and Zdk1. They tether Zdk1 to their cargo and LOV2 to sec61 (an ER localised membrane protein), which allows retention when not in the presence of 445nm light. They argue that their system demonstrates better kinetics and spatial specificity than the commonly used RUSH system. In addition, they argue it is more economically efficient and less toxic than the newer zapERtrap system. Using their new system, they demonstrate cargo trafficking through the Golgi apparatus and use Golgi ministacks to observe trafficking through the Golgi/RE. They then tested the effects of the Dynamin-2 inhibitor Dynasore and unexpectedly saw an earlier block than previously observed. Interestingly, they show that this is not due to the effects of blocking dynamin-2 as other inhibitors do not have the same effect, they thus conclude that there is an off target effect for some cis-Golgi uncoating, tethering, or fusion machinery. On the whole the manuscript is clear and well written and the core of the manuscript the

RudLOV system a valuable contribution to the field. The specific effect of the off-target by Dynasore is also interesting and novel; however, not as important as the system itself. I have some comments concerning some of the comparative measures, statistics and controls that I hope will improve this useful contribution.

In Figure 1, the kinetics of RUSH and RudLOV are compared. The authors conclude that the RUSH is less efficient. This may well be true; however, it is hard to draw that comparison as the comparison is not fair due to the use of different hooks. I think this is rather important as if this system is to be compared; it would be useful to do a fair comparison of the two systems.

In fact, RUSH uses KDEL and RudLOV uses Sec61 as a ER-retention signal for a hook. It is impossible to change the RudLOV hook to KDEL because the LOV2 C-terminal helix Ja must be free. The LOV2 C-terminal helix Ja consists of the interface recognized by Zdk1 and unwinds upon illumination. Thus, we rewrote the manuscript to the comparison between RUSH with streptavidin::KDEL hook and RudLOV with Sec61b::LOV2 hook.

It is also absolutely imperative to know what the time is between "pre" and "illumination"; if this is 10 mins, then the system is comparable to RUSH.

We used 10 min of illumination, and for clarity the illumination time is highlighted in blue in Figure 1G.

Although 10 minutes of illumination is not very short, the time to reach the peak of cargo at Golgi is shorter and more synchronized in RudLOV than in RUSH.

We also changed the way we numbered the minutes. We illuminated for 10 minutes and then started live imaging in our RudLOV experiments. In our original manuscript, we used time 0 when live imaging started. However, we have now used time 0 when illumination started. This makes the figure easier to understand (see comment for reviewer 3).

Finally, it is also essential to know the expression of the cargo. It is well known that cells overexpressing cargo RUSH with slower kinetics, and the cells in Panel D appear, to my eyes, to be overexpressing the cargo. This can be quantified with total GFP. If all these things are accounted for, this will be a much more useful comparison.

In addition, from my reading of the methods, the RudLOV system is stably integrated in the cells, while the RUSH system is transiently transfected. It is well known that transient transfections yield much higher expression, and also, the transfection itself affects trafficking- I think the authors need to be much more careful with the comparison or more careful with the conclusion. Really, this is the key to the manuscript, as at least being comparable to RUSH is enough to make this interesting.

We used the stable cell line expressing the hook (Sec61b::LOV2), but the cargoes were transiently expressed in RudLOV. Thus, the situation of RudLOV was comparable to that of RUSH. Thus, the temporal control of RudLOV is better than that of RUSH. In addition, RudLOV has the advantage of local activation and amount control of cargo release. In particular, we have added two graphs and original images to show RudLOV's ability to control the amount of cargo released (Figure 2D, EV2). We have described these graphs in detail in response to reviewer 1's comments.

It is not clear here how many independent repeats of this and all experiments have been performed, this is absolutely crucial for these sorts of observations.

We added the number of experiments we performed on Figure legends.

In addition, here and throughout the manuscript, the use of statistics is either not present or improper. As I said, this is the most important figure for the manuscript, in my opinion, so the findings need to be well-validated. In Figures 2, 3, and 4, cargo localisation is quantified through line scans. This has two issues: 1) the drawing of the line can affect the observation, and 2) it lacks statistical output. I think that to make these interpretable findings; it would be very useful to have proper statistical analysis here from multiple cells across multiple experiments.

We changed the line scans in Figure 3D–G, 4B–E and Figure EV3F,H,J,L to those showing the normalized means of 15 line profiles of Cargo, GM130 and GalT across the Golgi stack. As it is impossible to show the similar normalized means for Figure 2E,G,I,K,L, because these are the pattern of temporal localization of Cargoes.

The authors conclude that dynasore has an unexpected off-target effect. However, the concentration was not explored, and no positive control was used for the dynasore, thus, perhaps there is a problem with that batch of dynasore or that stock that causes the observed phenomenon.

We examined the effects of dynasore at 12, 25, 50 and 100 mM and added these images and the plots showing the normalized means of 15 line profiles of VSV-G, GM130 and GalT across the Golgi stack in Figure EV4E–L. The control of dynasore was shown in the original Figure EV11, which is now Figure EV4M. We have also added the following sentences in the Results and Discussions,
“We also investigated the dose-dependency of pre-administered dynasore in terms of inhibitory effects on cargo transport at the cis-Golgi cisternae and TGN. Pre-administration of 12 or 25 μ M dynasore inhibited the cargo exit from the TGN, but pre-administration of 50 or 100 μ M dynasore inhibited cargo movement at the cis-Golgi cisternae (Figure EV4E–L). The former is comparable to the IC50 value of dynasore to

dynamamin-1/2, but the latter is higher. To confirm dynamamin-2 inhibition, we confirmed that 100 μ M dynasore inhibited the uptake of transferrin (Figure EV4M).

I find the speculation of the actual molecular target of the off-target effect lacking evidence, and although I leave it to the authors, without evidence, it is perhaps better not to make a claim on the target.

We'd like to retain these discussions.

The statistical comparison in Figure 4F is not valid. It is a student's T-test across a time series, which is not a proper use of that test. I advise the authors to get advice on the proper statistical test to perform here and throughout.

In the previous manuscript, the descriptions about the statistics were incorrect. Actually, to make the Figure 4F, we have performed two-tailed unpaired Welch's t-test for each pair of samples (control vs pre, etc), individually for each timepoint.

We fixed the figure legends and more description to clarify what we did as follow.

*"Results of two-tailed unpaired Welch's t-test for each combination of samples are shown in rows on top, as *** $P < 0.001$; ** $P < 0.01$; * $P < 0.05$."*

We also added the following sentences in Material and Methods,

"Briefly, for each timepoint, slices in the Z-stack were concatenated into a montage, regions positive to GM130, GalT and Cargo were defined, and the Mander's overlap coefficient (MOC) were calculated for each montage."

Referee #3:

In this study, Tago et al describe a new system of controlled cargo release within the secretory pathway and use it to demonstrate that treatment of cells with the dynamamin inhibitor dynasore impacts two steps in the secretory pathway - cis/medial Golgi maturation and TGN exit. They then show that only the latter is affected by other dynamamin inhibitors and thus dynasore is likely targeting a different protein to regulate cisternal maturation. Unfortunately, this target remains unidentified, but this finding serves as a warning to dynasore users.

The behavior of their new RudLOV2 system for controlled cargo release seems robust and has superior spatial resolution over the current predominant alternative, RUSH. It does however appear to have limitations as the illumination period is restrictive for imaging early ER-Golgi dynamics and it cannot be

used with certain experimental set ups - the authors themselves revert to using RUSH for some experiments. Nonetheless it may be of use to some in the membrane trafficking field looking to diversify their toolset.

It is true that RudLOV needs a relatively long light exposure, probably because light-form of LOV2 quickly returns to dark-form spontaneously.

It is worth noting that other photoactivatable transport systems have already been reported, for example UVR8 from the Kennedy group. It is not clear from this manuscript what advantage RudLOV2 has over these comparable systems and this should be discussed in the manuscript.

We added the following sentences in introduction,

“Light-triggered protein secretion systems allow for more precise control of cargo secretion. The first such system used a plant photoreceptor protein, UVR8, which forms a photolabile homodimer (Chen et al., 2013). Multiple UVR8-fused cargo proteins are sequestered in the ER, and a brief pulse of light triggers forward cargo trafficking through the secretory pathway to the plasma membrane. The rapid release of the cargo from the ER is robust, but the use of UV light (~310 nm) can damage cells. The same group recently developed another light-induced system, the zapalog-mediated ER trap (Bourke et al., 2021), in which milder 405-nm light illumination is used.”

We also added the following sentences in conclusion,

“In addition, the use of the relatively longer wavelengths of light (445–488nm) reduces damage to the cells compared with the previously reported UVR8-based light trigger method (310 nm) or zapalog-mediated ER trap (405 nm) (Bourke et al., 2021; Chen et al., 2013).”

Whilst the majority of the data is well presented, there are two main areas I believe require improvement to strengthen the premise and conclusions of this study:

1) With respect to the RUSH experiments, I do not think the data, as currently presented, support the conclusion that higher expression levels result in delayed cargo export as reported in line 107-108. The quantification provided only shows time of enrichment in the Golgi, without any correlation to expression levels. Plotting time against signal intensity for example would better represent this conclusion.

We plotted the total cargo signal intensity in pre-illuminated cells on Y-axis and the time in which the cargo signal intensity become maximum in Golgi stack on X-axis in Figure EV1B. This graph clearly shows the cells with higher expression of cargos delay the cargo exit from ER.

I am also concerned that the VSVG-RUSH construct appears to be leaky - the VSVG signal looks to be in the Golgi region before treatment in most cells, and so it is unclear which cells they have chosen to quantify and how this will impact the dynamics. There are other RUSH cargoes which are less leaky that the authors could use - and they seem to have TNFa already - or they could increase the ratio of hook:cargo by expressing another hook vector or changing the promoters. Alternatively, as this data is just to set up the premise of this study, there is a lot of literature reporting the dynamics of different cargoes which could be cited to back up this data set.

We believe that the VSVG-RUSH construct is not leaky. The figure shows the localization of VSVG-RUSH construct on ERES, which are abundant near the Golgi ribbon. ERES accumulation of TNFa prior to biotin administration has been reported, and we also showed that TNFa-RUSH construct localizes to ERES prior to biotin administration in Figure 2C in the original manuscript (Weigel et al., 2021). The VSVG-RUSH construct behaves similarly. We have added the images showing VSVG colocalization with the ERES marker Sec13 before biotin administration (Figure EV1C). We also detected VSVG on ERES 10 min after the initial detection without biotin, indicating that this VSVG did not enter the Golgi stacks. Thus, this VSVG-RUSH construct is not leaky.

Finally, according to the materials and methods section, GalT:iRFP713 was co-transfected with these cells. Inclusion of this marker in the figure would help the reader relate cargo location and Golgi enrichment.

As we think the signal of cargo is clearer without GalT, we used cargo only picture in Figure 1A, F, but we added the double-colored pictures with GalT in Figure EV1A, D.

2) In lines 222-225 the authors suggest that dynasore 'inhibits the uncoating, tethering or fusion of COPI vesicles' on the basis that there are eCOP-APEX2 labelled vesicles in micrographs of cells treated with dynasore. Whilst this is a good hypothesis that fits with their model, this evidence is not compelling, mainly due to the lack of controls. eCOP-APEX2 would be expected to label vesicles in untreated cells and so it is not remarkable that they are seen here. The authors need untreated control samples and some quantification to show there is an unusual build-up of COPI vesicles in the dynasore treated cells to support their conclusions. It might also be possible to quantify COPI punctae by immunofluorescence depending on how tightly they pack to the peri-Golgi region.

Evidence of vesicle accumulation is also important to the authors model that dynasore is inhibiting a protein required for COPI-dependent cis/medial Golgi maturation, whereas other dynamin inhibitors do not.

As we have the serial sections containing more than 5 whole Golgi stacks in both dynasore treated and untreated cells, we counted the number of all vesicles for each of whole Golgi stacks. 2.5 times of vesicles are found in dynasore treated Golgi compared to untreated Golgi. This result is included in Figure EV5C, and the following sentences is added in Result and Discussion.

“We counted the number of vesicles near Golgi stacks in dynasore treated and untreated cells (Figure EV5C) and identified 810.2 (\pm 248.2) and 326.8 (\pm 83.4) vesicles near a Golgi stack, respectively.”

It would therefore also be worth looking at dyngo4a or dynole 34-2 treated cells by EM too to see whether this also generates a build-up of vesicles. If they don't, this is the step specifically affected by dynasore in the early secretory pathway, and if they do, it is the result of dynamin inhibition at the TGN and not the dynasore-specific step.

In this revised manuscript, we observed Golgi stacks in dynole NC (control), dynole, and Dyngo4a treated cells by TEM (Figure EV5E–G), and the number of vesicles near Golgi stacks in a single section in dynole NC (control), dynasore, dynole, and Dyngo4a treated cells are counted and plotted (Figure EV5H). The number of vesicles near Golgi stacks was significantly increased in dynasore treated cells compared to those in dynole NC (control) treated cells, but not in dynole and Dyngo4a treated cells. Since these cells expressed eCOP-APEX2, COPI-coated vesicles are easily identified. However, it is not clear that COPI-coated vesicles are increased in dynasore treated cells, so we could not conclude whether dynasore inhibits uncoating or subsequent steps.

We added the following sentences in Result and Discussion.

“We observed dynole, dynole NC (control), and dyngo4a treated cells with eCOP-APEX2 by transmission electron microscopy (TEM; Figure EV5D–G). Some COPI vesicles were observed in these Golgi stacks, but the average number of vesicles counted in single TEM sections was increased only in dynasore treated cells (Figure EV5H).”

Minor points:

1) The authors do not define what LOV2 is. This should be included in the introduction

We added the following sentence in the introduction.

*“LOVTRAP utilizes two proteins: (1) LOV2 (light-oxygen-voltage), a photo-sensor domain from *Avena sativa* phototropin 1 and (2) a small artificial protein, Zdk1, which binds selectively to the dark state of LOV2.”*

2) I would suggest that Figure1 be rearranged so that the data appears in the same order as mentioned in the text - the RUSH data splits up the RudLOV story at the moment.

We rearranged Figure 1A-E.

3) Figure 1G is not referred to in the results/discussion.

We added the following sentences in the result/discussion.

“In 53% of cells, cargo amount on Golgi peaked within 20 min after biotin administration, but others were not (Figure 1B).”

“In 96% of cells, cargo amount on Golgi peaked within 20 min after start of illumination (Figure 1G).”

4) Figure 1G-I As currently displayed, it is a little confusing to call the first time point after illumination 0min - it looks like translocation is instantaneous. Perhaps this could be labelled differently, have a bar indicating light and dark, or at least in the figure legend state the length of illumination time.

We used 10 minutes of illumination and then started live-imaging immediately after the illumination is done.

In our original manuscript, we used the label "time 0" at the time when live-imaging was started. However, as the reviewer points out, this is confusing. In our revised manuscript, we used "time 0" at the onset of illumination. Thus, the time when the observation started is now 10 minutes. We have changed the figure, legend and text according to this definition.

5) Figure 2D-J To better follow the arguments made in the text about the journey of each protein, linescans at multiple relevant timepoints would be more helpful than just the end point. The authors should also be aware that the images in these figures will be uninterpretable to readers with red-green colour blindness and should consider using different colour combinations eg magenta not red.

We added line plots and the two-color images with green and magenta in Appendix Figure S1, S2.

6) Figure 4G-K are not referenced in the text.

We fixed

7) The authors do not indicate how many times experiments were repeated as so it is difficult to assess how robust their findings are. This should be included in the figure legends, especially for those which were quantified and statistically analysed.

We added the number of experiments we performed on Figure legends.

Dear Dr. Satoh,

Thank you for submitting your revised manuscript. It has now been seen by all of the original referees.

As you can see, the referees find that the study is significantly improved during revision and recommend publication. However, I need you to address the points below before I can accept the manuscript.

- Please address the remaining concern of referee #2.
- Please rename the 'Competing Interests' section as "Disclosure Statement and Competing Interests" and move it after the Acknowledgements section.
- Please remove 'Author Contributions' section from the manuscript text.
- Please fill out and include an author checklist as listed in our online guidelines (<https://www.embopress.org/page/journal/14693178/authorguide>).
- We note that the funding information is currently not congruent; the following is missing as separate entries in our manuscript tracking system:
 - CREST grant JPMJSP2132
 - Core Research for Organellar Diseases funding from Hiroshima University, Takeda Science Foundation, Ohsumi Frontier Science Foundation
 - Grant-in-Aid for Transformative Research Areas (Platforms for Advanced Technologies and Research Resources "Advanced Bioimaging Support" grant no. JP22H04926)
- We note that Figures 2J, N and R are currently not called out in the text.
- Supplemental Table 1, also titled Appendix Table S1, should be renamed to Dataset EV1 and the legend should be removed from the Appendix file and provided in the Excel (as a separate sheet/tab). The callout in the text should be updated accordingly.
- The appendix file needs to be submitted in PDF format and needs a Table of Contents with page numbers; the callouts in the ms need to be corrected to Appendix Figure S1, Appendix Figure S2; the titles in the figures themselves need to be updated accordingly (currently Figure S1, Figure S2, which is incorrect).
- We note that a video 1 is called out and a legend is provided in the manuscript, but the actual video is missing; the legend should be removed from the ms and provided in a readme.txt file; the movie file and the legend should be zipped up together and uploaded as one zip folder; the correct source file names, titles and ms callouts should be Movie EV1.
- All research articles submitted as revised versions must include a structured methods section that includes a Reagents and Tools Table followed by a Methods and Protocols section. Please see <https://www.embopress.org/page/journal/14693178/authorguide#structuredmethods> for further information.
- Please rename Materials and Methods as Methods.
- We noted a potential Figure re-use between Figure 3M and Figure EV5D. Image is rotated. Please clarify, also in the legends of both panels.
- Please make the dataset deposited in Dryad (10.5061/dryad.jm63xsjhs) publicly available and provide a link that directly resolves to the dataset in the Data Availability section.
- Our production/data editors have asked you to clarify several points in the figure legends:
 - o Please note that the exact p values are not provided in the legends of figures 4f, j; EV 5c, h.
 - o Please note that information related to n is missing in the legends of figures EV 4f, h, j, l.
 - o Please note that the error bars are not defined in the legends of figures 2d; EV 4f, h, j, l.
 - o Please note that the measure of center for the error bars needs to be defined in the legends of figures 4f, j; EV 5c, h.
 - o Please note that scale bar and its definition are missing for figures 2f-t; 3d-g; 4b-e; EV 1c; EV 4b, d.
 - o Please note that the white arrows are not defined in the legend of figure EV 4b, d. This needs to be rectified.
- Papers published in EMBO Reports include a 'synopsis' and 'bullet points' to further enhance discoverability. Both are displayed on the html version of the paper and are freely accessible to all readers. The synopsis includes a short standfirst summarizing the study in 1 or 2 sentences (max 35 words) that summarize the paper and are provided by the authors and streamlined by the handling editor. I would therefore ask you to include your synopsis blurb and 3-5 bullet points listing the key experimental findings.
- In addition, please provide an image for the synopsis. This image should provide a rapid overview of the question addressed in the study but still needs to be kept fairly modest since the image size cannot exceed 550 (width) x 300-600 (height) pixels.

Thank you again for giving us to consider your manuscript for EMBO Reports, I look forward to your minor revision.

Kind regards,

Deniz Senyilmaz Tiebe

--

Deniz Senyilmaz Tiebe, PhD

Referee #1:

The authors have addressed my concerns with this revised manuscript. I enthusiastically recommend publication.

Referee #2:

Tago et al. have resubmitted their manuscript in which they present a novel tool for the synchronous trafficking of cargo through the secretory and endolysosomal systems. They have responded to each reviewer's comments and clarified several concerns. I particularly appreciate the additions of the new data on the temporal control of their system and the new temporal numbering (starting from the start of illumination), which I think improves the manuscript's clarity for the reviewer.

I remain concerned about the use of quantification and statistics. Although I appreciate the clarifications on the number of repeats and the statistical tests, which were incorrect in the initial submission, I understand that using a two-sample statistic is inappropriate with multiple samples without a post hoc correction. As I stated in my initial review, I am not a statistician. However, it was my understanding that this can lead to an increased risk of a type-1 error.

Referee #3:

The authors have been very thorough in addressing all the reviewers comments experimentally as well as in the manuscript. I believe their conclusions seem greatly strengthened and the study is ready for publication.

Dear Dr. Satoh,

Thank you for submitting your revised manuscript. It has now been seen by all of the original referees.

As you can see, the referees find that the study is significantly improved during revision and recommend publication. However, I need you to address the points below before I can accept the manuscript.

- Please address the remaining concern of referee #2.

Referee#2

I remain concerned about the use of quantification and statistics. Although I appreciate the clarifications on the number of repeats and the statistical tests, which were incorrect in the initial submission, I understand that using a two-sample statistic is inappropriate with multiple samples without a post hoc correction. As I stated in my initial review, I am not a statistician. However, it was my understanding that this can lead to an increased risk of a type-1 error.

Thank you for your insightful comments. Based on your suggestion, we reanalyzed the data using the Steel-Dwass test. While we observed a few type I errors, the overall results remained unchanged.

- Please rename the 'Competing Interests' section as "Disclosure Statement and Competing Interests" and move it after the Acknowledgements section.

We fixed.

- Please remove 'Author Contributions' section from the manuscript text.

We fixed.

- Please fill out and include an author checklist as listed in our online guidelines (<https://www.embopress.org/page/journal/14693178/authorguide>).

We filled out the checklist.

- We note that the funding information is currently not congruent; the following is missing as separate entries in our manuscript tracking system:

- SPRING grant JPMJSP2132 (tago spring)
- Core Research for Organelle Diseases funding from Hiroshima University, Takeda Science Foundation,
- Ohsumi Frontier Science Foundation
- Grant-in-Aid for Transformative Research Areas (Platforms for Advanced Technologies and Research Resources "Advanced Bioimaging Support" grant no. JP22H04926)

We fixed.

- We note that Figures 2J, N and R are currently not called out in the text.

We fixed.

- Supplemental Table 1, also titled Appendix Table S1, should be renamed to Dataset EV1 and the legend should be removed from the Appendix file and provided in the Excel (as a separate sheet/tab). The callout

in the text should be updated accordingly.

We fixed.

- The appendix file needs to be submitted in PDF format and needs a Table of Contents with page numbers; the callouts in the ms need to be corrected to Appendix Figure S1, Appendix Figure S2; the titles in the figures themselves need to be updated accordingly (currently Figure S1, Figure S2, which is incorrect).

We fixed.

- We note that a video 1 is called out and a legend is provided in the manuscript, but the actual video is missing; the legend should be removed from the ms and provided in a readme.txt file; the movie file and the legend should be zipped up together and uploaded as one zip folder; the correct source file names, titles and ms callouts should be Movie EV1.

We fixed.

- All research articles submitted as revised versions must include a structured methods section that includes a Reagents and Tools Table followed by a Methods and Protocols section. Please see <https://www.embopress.org/page/journal/14693178/authorguide#structuredmethods> for further information.

We included Reagents and Tools Table.

- Please rename Materials and Methods as Methods.

We fixed.

- We noted a potential Figure re-use between Figure 3M and Figure EV5D. Image is rotated. Please clarify, also in the legends of both panels.

Thanks for the note. Actually, Figure EV5D is supposed to repeat Figure 3M to allow side-by-side comparison with new experiments with different drug treatment (Figure EV5E-G). We have replaced Figure EV5D with another image taken from a different sample.

- Please make the dataset deposited in Dryad (10.5061/dryad.jm63xsjhs) publicly available and provide a link that directly resolves to the dataset in the Data Availability section.

We made dataset available.

- Our production/data editors have asked you to clarify several points in the figure legends:
 - o Please note that the exact p values are not provided in the legends of figures 4f, j; EV 5c, h.

We provided p-values for EV 5c, h.

- o Please note that information related to n is missing in the legends of figures EV 4f, h, j, l.

We fixed.

- o Please note that the error bars are not defined in the legends of figures 2d; EV 4f, h, j, l.

We added the following sentence.

“Error bars are presented as mean \pm SD.”

- o Please note that the measure of center for the error bars needs to be defined in the legends of figures 4f, j; EV 5c, h.

We added the following sentence.

“Error bars are presented as mean \pm SD.”

o Please note that scale bar and its definition are missing for figures 2f-t; 3d-g; 4b-e; EV 1c; EV 4b, d.

As indicated in the original manuscript, we have used arrows to show a length of 1.5 μ m instead of a scale bar.

o Please note that the white arrows are not defined in the legend of figure EV 4b, d. This needs to be rectified.

In the original manuscript, we defined as the following.

“Signal intensity was measured along the arrow (representing 3 μ m).”

- Papers published in EMBO Reports include a 'synopsis' and 'bullet points' to further enhance discoverability. Both are displayed on the html version of the paper and are freely accessible to all readers. The synopsis includes a short standfirst summarizing the study in 1 or 2 sentences (max 35 words) that summarize the paper and are provided by the authors and streamlined by the handling editor. I would therefore ask you to include your synopsis blurb and 3-5 bullet points listing the key experimental findings.
- In addition, please provide an image for the synopsis. This image should provide a rapid overview of the question addressed in the study but still needs to be kept fairly modest since the image size cannot exceed 550 (width) x 300-600 (height) pixels.

'synopsis' is the following.

“This study developed a new cargo releasing method, RudLOV, which has three exceptional control abilities: spatial, temporal, and quantitative control of cargo-release. RudLOV revealed non-canonical effects of the well-known dynamin inhibitor dynasore on early-Golgi transport.”

'bullet points' is the following.

- Spatially restricted cargo release by RudLOV allowed observation of local transport within the cells.
- Precise temporal control of cargo release by RudLOV successfully dissected effects of dynasore on early and late stages of intra-Golgi transport.
- Quantitative control of cargo release by RudLOV visualized the cisternal movement of the cargos and their specific exit sites.
- Cargos can be released by 445 or 488 nm laser, with less damage than by UV.
- RudLOV can work without exogenous chemicals. This simplicity facilitates *in situ* application in animal cells.

Dr. Akiko Satoh
Hiroshima University
Division of Life Science, Graduate School of Integral Arts and Science,
1-7-1, Kagamiyama
Higashi-hiroshima 739-8521
Japan

Dear Dr. Satoh,

Thank you for submitting your revised manuscript. I have now looked at everything and all is fine. Therefore, I am very pleased to accept your manuscript for publication in EMBO Reports.

Congratulations on a nice work!

Before exporting your manuscript to our production team, I need your input on one more point. I made some minor changes in the below items. Please take a look and confirm, or feel free to propose further changes. Thank you.

Title:

RudLOV is a new optically synchronized cargo transport method revealing unexpected effects of dynasore

Abstract:

Live imaging of secretory cargoes is a powerful method for understanding the mechanisms of membrane trafficking. Inducing the synchronous release of cargoes from an organelle is key for enhancing microscopic observation. We developed an optical cargo-releasing method, 'retention using dark state of LOV2' (RudLOV), which enables precise spatial, temporal, and quantity control during cargo release. A limited amount of cargo-release using RudLOV is able to visualize cargo cisternal-movement and cargo-specific exit sites on the Golgi/trans-Golgi network. Moreover, by controlling the timing of cargo-release using RudLOV, we reveal the canonical and non-canonical effects of the well-known dynamin inhibitor dynasore, which inhibits early- but not late-Golgi transport and exits from the trans-Golgi network where dynamin-2 is active. Accumulation of COPI vesicles at the cis-side of the Golgi stacks in dynasore-treated cells suggests that dynasore targets COPI-uncoating/tethering/fusion machinery in the early-Golgi cisternae or endoplasmic reticulum but not in the late-Golgi cisternae. These results provide insight into the cisternal maturation of Golgi stacks.

Kind regards,

Deniz Senyilmaz Tiebe

--

Deniz Senyilmaz Tiebe, PhD
Senior Scientific Editor
EMBO Reports
